# Cullin3-KLHL15 ubiquitin ligase mediates CtIP protein turnover to fine-tune DNA-end resection

Lorenza P. Ferretti[1], Sarah-Felicitas Himmels[1], Anika Trenner[1], Christina Walker[1], Christine von Aesch[1], Aline Eggenschwiler[1], Olga Murina[1], Radoslav I. Enchev[2], Matthias Peter[2], Raimundo Freire[3], Antonio Porro[1] & Alessandro A. Sartori[1]

Human CtIP is a decisive factor in DNA double-strand break repair pathway choice by enabling DNA-end resection, the first step that differentiates homologous recombination (HR) from non-homologous end-joining (NHEJ). To coordinate appropriate and timely execution of DNA-end resection, CtIP function is tightly controlled by multiple protein–protein interactions and post-translational modifications. Here, we identify the Cullin3 E3 ligase substrate adaptor Kelch-like protein 15 (KLHL15) as a new interaction partner of CtIP and show that KLHL15 promotes CtIP protein turnover via the ubiquitin-proteasome pathway. A tripeptide motif (FRY) conserved across vertebrate CtIP proteins is essential for KLHL15-binding; its mutation blocks KLHL15-dependent CtIP ubiquitination and degradation. Consequently, DNA-end resection is strongly attenuated in cells overexpressing KLHL15 but amplified in cells either expressing a CtIP-FRY mutant or lacking KLHL15, thus impacting the balance between HR and NHEJ. Collectively, our findings underline the key importance and high complexity of CtIP modulation for genome integrity.

[1] University of Zurich, Institute of Molecular Cancer Research, Winterthurerstrasse 190, 8057 Zurich, Switzerland. [2] ETH Zurich, Institute of Biochemistry, Department of Biology, Otto-Stern-Weg 3, 8093 Zurich, Switzerland. [3] Unidad de Investigación, Hospital Universitario de Canarias, Instituto de Tecnologías Biomédicas, Ofra s/n, La Cuesta, La Laguna, Tenerife, Spain. Correspondence and requests for materials should be addressed to A.A.S. (email: sartori@imcr.uzh.ch).

To preserve genome integrity, cells have evolved a complex system of DNA damage detection, signalling and repair: the DNA damage response (DDR). Following genotoxic insults, upstream DDR factors rapidly assemble at damaged chromatin, where they activate lesion-specific DNA repair pathways as well as checkpoints to delay cell cycle progression, or, if DNA repair fails, to trigger apoptosis[1]. DNA double-strand breaks (DSBs) are one of the most lethal types of DNA damage with the potential to cause genomic instability, a hallmark and enabling characteristic of cancer[2]. DSBs are induced by ionizing irradiation (IR) or frequently arise during replication when forks collide with persistent single-strand breaks, such as those generated by camptothecin (CPT), a DNA topoisomerase I inhibitor[3]. To maintain genome stability, cells have evolved two major pathways dealing with the repair of DSBs: non-homologous end-joining (NHEJ) and homologous recombination (HR)[4]. NHEJ is the canonical pathway during G0/G1 phase of the cell cycle and repairs the majority of IR-induced DSBs. In this process, broken DNA ends are religated regardless of sequence homology, making NHEJ potentially mutagenic[5]. HR, instead, is an error-free repair pathway, which requires the presence of an undamaged homologous template, usually the sister chromatid[6]. Thus, HR is restricted to S and G2 phases of the cell cycle and preferentially repairs DSBs resulting from replication fork collapse[7]. The first step of HR, termed DNA-end resection, involves the processing of one DSB end to generate 3′ single-stranded DNA (ssDNA) tails that, after being coated by the Rad51 recombinase, mediate homology search and invasion into the sister chromatid strand. DNA-end resection is initiated by the combined action of the MRE11–RAD50–NBS1 (MRN) complex and CtIP[8], and is a key determinant of DSB repair pathway choice, as it commits cells to HR by preventing NHEJ[9].

The ubiquitination and neddylation machineries have recently emerged as a crucial players for maintaining genome stability by orchestrating key DDR events including various DNA repair pathways[10,11]. Ubiquitination of target proteins involves the concerted action of three factors: E1 ubiquitin-activating enzymes, E2 ubiquitin-conjugating enzymes and E3 ubiquitin ligases, which determine substrate specificity[12]. Among the estimated > 600 human E3s, Cullin-RING ligases (CRLs) are the most prevalent class, controlling a plethora of biological processes[13,14]. Although few CRLs, in particular those built up by Cullin1 (also called SCF complex) and Cullin4, were shown to function in cell cycle checkpoint control and nucleotide excision repair[15], a role for CRLs in the regulation of DSB repair has so far remained largely elusive.

Here, we identify the human Kelch-like protein 15 (KLHL15), a substrate-specific adaptor for Cullin3 (CUL3)-based E3 ubiquitin ligases, as a novel CtIP interaction partner. We show that CUL3-KLHL15 catalyses polyubiquitination and proteasome-dependent degradation of CtIP. Mechanistically, we find that KLHL15 recognizes a short-tripeptide motif (FRY) located in the conserved C-terminal domain (CTD) of CtIP and that mutation of this motif protects CtIP from KLHL15-dependent degradation resulting in prolonged CtIP protein half-life and, consequently, excessive DNA-end resection. We further demonstrate that cells lacking KLHL15 phenocopy the behaviour of FRY mutant cells, including reduced NHEJ efficiency as a consequence of CtIP protein accumulation and increased resection. Finally, we provide evidence that PIN1-dependent isomerization of CtIP facilitates its targeting by CUL3-KLHL15. Taken together, our results uncover a critical role for CUL3-KLHL15 ubiquitin ligase in governing CtIP DNA-end resection activity and DSB repair pathway choice.

## Results

**CtIP interacts with the CUL3 substrate adaptor KLHL15.** To gain further insights into the regulation of DNA-end resection in human cells, we searched for novel interacting partners of CtIP by tandem affinity purification followed by mass spectrometry. To this end, we generated a stable HEK293 cell line inducibly expressing Strep-HA-tagged CtIP. To reduce the number of false positive protein–protein interactions, we used a tetracycline concentration (10 ng ml$^{-1}$) at which the expression levels were comparable to those of endogenous CtIP (Supplementary Fig. 1a). Remarkably, besides the CtIP bait protein, only two proteins were identified with multiple peptides in two independent mass spectrometry analyses, Kelch-like protein 15 (KLHL15) and Cullin3 (CUL3) (Fig. 1a; and Supplementary Fig. 1b). KLHL15 belongs to a family of 42 human proteins that possess an N-terminal BTB-BACK domain and a C-terminal Kelch domain comprising five to six Kelch repeats[16]. Kelch-like proteins bind CUL3 through their BTB-BACK domain, whereas substrate recognition is mediated by the Kelch domain[17–19]. Importantly, many Kelch-like proteins function as substrate-specific adaptors in CUL3-based E3 ligase complexes to catalyse protein ubiquitination and degradation by the 26S proteasome[20].

To confirm the result of our proteomic analysis, suggesting that CtIP and CUL3-KLHL15 form a complex, we first performed an immunoprecipitation (IP) experiment and found that endogenous KLHL15 and CUL3 efficiently co-precipitated with GFP-tagged CtIP (Fig. 1b). Similarly, endogenous CtIP was successfully recovered following co-IP of FLAG-tagged KLHL15, but was not present in co-immunocomplexes containing CUL3 and KLHL22 (Supplementary Fig. 1c), which is the closest ortholog of KLHL15 (refs 18,21). Next, we generated two FLAG-KLHL15 truncation mutants lacking either the Kelch (ΔKelch) or the BTB-BACK (ΔB + B) domain (Fig. 1c,d). As expected, CUL3-KLHL15 interaction was largely abolished in the ΔB + B mutant, whereas CtIP binding to KLHL15 was strongly impaired in the ΔKelch mutant (Fig. 1d). Interestingly, we observed that direct interactions between the individual KLHL15 domains and CUL3 or CtIP were weakened in both truncation mutants, indicating that the structural integrity of full-length KLHL15 is important for stable ternary complex formation between CtIP, KLHL15 and CUL3 (Fig. 1d). In case of the ΔB + B mutant, this could in part be explained by the fact that the BTB adaptor domain was shown to be required for the efficient dimerization and activation of CUL3 ubiquitin ligase complexes[18,22,23]. To identify specific amino acid residues in KLHL15 that mediate binding to CUL3, we performed multiple sequence alignments of various human BTB-Kelch proteins and the MATH-BTB protein SPOP. This revealed the presence of a conserved, paired helix structure following the BTB domain of KLHL15, which is predicted to constitute a CUL3-interacting box (3-box) (Fig. 1c and Supplementary Fig. 1d)[23]. Similar to what we observed for the ΔB + B truncation mutant, substituting either asparagine 132 or isoleucine 136 for alanine (N132A and I136A, respectively) in the 3-box of full-length FLAG-KLHL15 decreased its interaction with CUL3, especially with the neddylated form of CUL3 (Fig. 1c,e). Neddylation, the covalent attachment of the small ubiquitin-like protein NEDD8 to proteins, has been demonstrated to be essential for the activation of most cullin-based E3 ligases[13,24]. Next, through sequence alignments of the Kelch domain of human and zebrafish KLHL15 with the respective Keap1 (alias KLHL19) homologues, we located glycine 386 in the third repeat (G386) and tyrosine 552 (Y552) in the sixth repeat (Fig. 1c and Supplementary Fig. 1e). Importantly, mutation of these residues in Keap1 disrupted its ability to bind Nrf2 and repress Nrf2-dependent transcription[25,26]. Consistently,

we observed that KLHL15-G386C and -Y552A mutants were defective in CtIP binding (Fig. 1c,f).

In the course of these experiments, we noticed that endogenous CtIP protein levels were differentially altered in cells transfected with KLHL15 expression constructs with a marked decrease in presence of wild-type (wt) KLHL15 (Fig. 1d–f). To further address this issue, we measured CtIP protein stability using a cycloheximide (CHX) chase assay and found that CtIP shows a decreased protein half-life and is more rapidly degraded in cells overexpressing KLHL15-wt than in cells transfected with KLHL15-Y552A mutant (Supplementary Fig. 1f). Taken together, these results demonstrated that CUL3-KLHL15 specifically interacts with CtIP and controls its protein turnover.

**CRL3[KLHL15] ubiquitin ligase targets CtIP for degradation**. To confirm our above results, using a different cell system, we generated stable U2OS cells expressing doxycycline (Dox)-inducible GFP-tagged KLHL15. As shown in Fig. 2a, we detected a significant decrease of endogenous CtIP protein levels already at 24 h following induction of KLHL15 expression and found that this was partially restored by addition of the proteasome inhibitor MG-132. Similarly, treatment of KLHL15-induced cells with MLN-4924, a small molecule inhibitor of the NEDD8-activating enzyme[27], greatly increased CtIP protein stability, indicating that CtIP degradation by KLHL15 requires CUL3 activation (Fig. 2b). Specifically, our results indicated that CUL3 acts together with

KLHL15 in promoting CtIP degradation, as its downregulation stabilized CtIP in cells overexpressing KLHL15 (Fig. 2c). Consistently, transfection of three siRNA oligos targeting different regions in the KLHL15 transcript reproducibly increased CtIP protein levels in parental U2OS cells (Fig. 2d and Supplementary Fig. 2a); without affecting cell cycle distribution (Supplementary Fig. 2b). Of note, we only detected a robust KLHL15 protein signal in the chromatin-enriched fraction, suggesting a role for KLHL15 in targeting substrates predominantly in the nucleus (Fig. 2d). Furthermore, using stable U2OS[GFP-KLHL15] cells, siRNA oligos targeting the coding sequence of KLHL15 (#1 and #2) efficiently depleted exogenous KLHL15, thereby restoring CtIP protein levels (Fig. 2e). In contrast, CtIP was still degraded upon induction of GFP-KLHL15 in cells transfected with siRNA against the 3′-untranslated region of KLHL15 (#3) specifically silencing endogenously expressed KLHL15 (Fig. 2e). Consistent with a key function of KLHL15 in regulating CtIP protein turnover, we observed that the half-life of CtIP was prolonged after KLHL15 downregulation (Supplementary Fig. 2c).

Proteasome-dependent proteolysis typically requires the conjugation of ubiquitin to the target protein. Therefore, we next addressed whether KLHL15 mediates CtIP ubiquitination *in vivo*. To this end, we transfected HEK293 cells inducibly expressing GFP-CtIP with histidine-tagged ubiquitin and analysed the level of CtIP ubiquitination after Ni-NTA pull-down under denaturing conditions. As previously reported[28,29], we readily detected

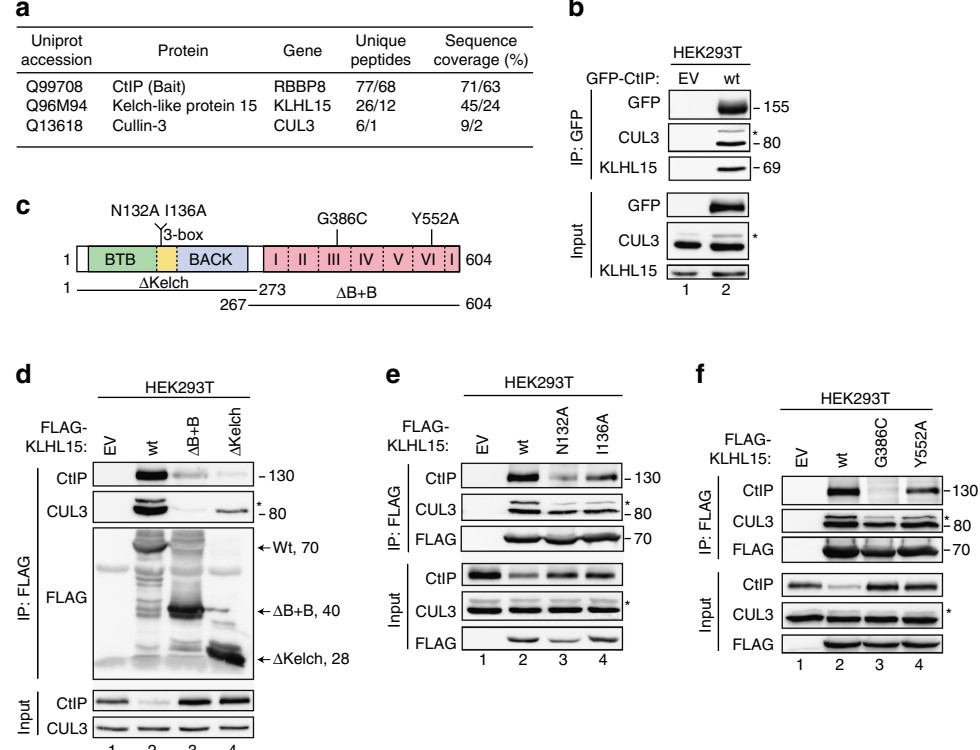

**Figure 1 | CtIP interacts with the CUL3-KLHL15 complex.** (**a**) HEK293 cells inducibly expressing StrepHA-tagged CtIP were used for tandem affinity purification of protein complexes. The number of unique peptides and sequence coverage for the proteins identified by mass spectrometry are listed. '/' delimitates the data from two biological replicates. (**b**) HEK293T cells were transfected with either empty vector (EV) or the GFP-CtIP expression constructs. Forty-eight hours after transfection, cells were lysed and whole-cell extracts were subjected to IP using anti-GFP affinity resin. Inputs and recovered protein complexes were analysed by immunoblotting. (**c**) Schematic representation of the human KLHL15 protein indicating truncation and single-amino acid point mutants thereof used in **d**–**f**. '3-box' denotes CUL3-interacting box, whereas the six Kelch repeats are indicated in pink. (**d**–**f**) HEK293T cells were transfected with either EV or the indicated FLAG-KLHL15 expression constructs. Forty-eight hours after transfection, cells were lysed and whole-cell extracts were subjected to IP using anti-FLAG M2 affinity resin. Inputs and recovered protein complexes were analysed by immunoblotting. Asterisks indicate neddylated CUL3.

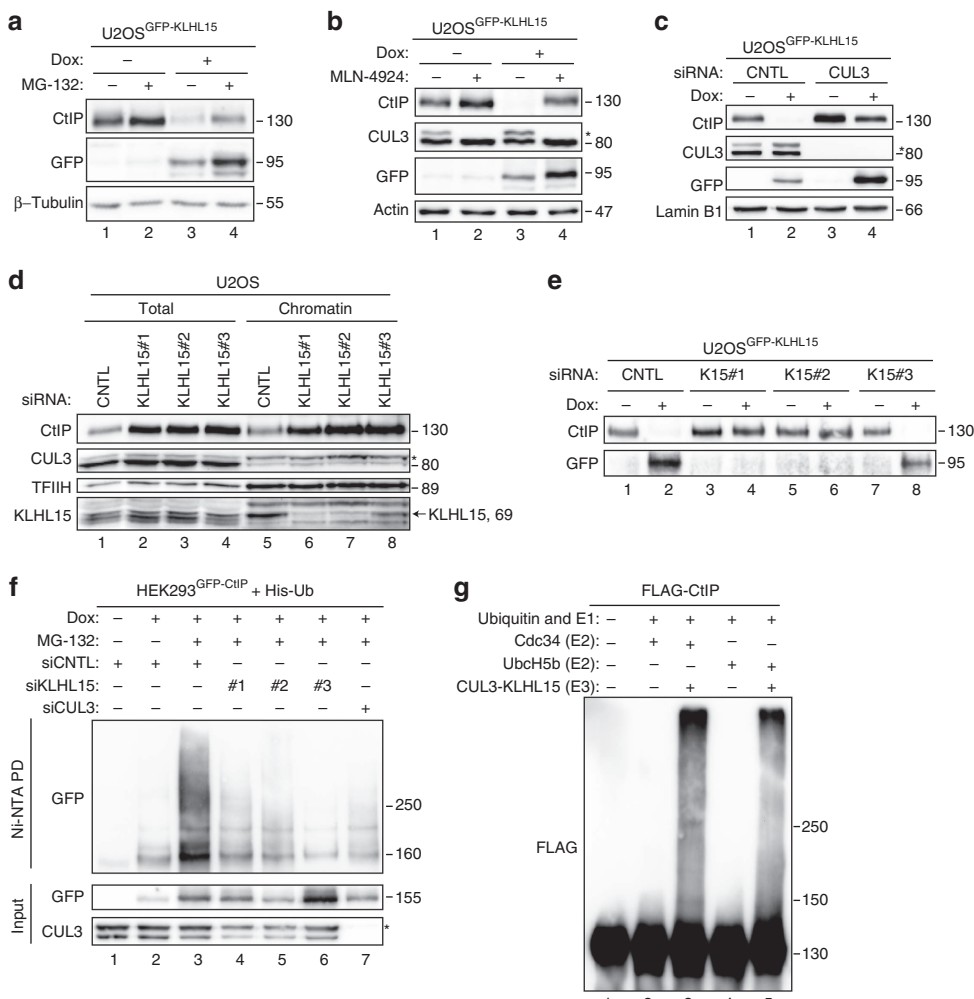

**Figure 2 | CUL3-KLHL15 E3 ligase promotes CtIP protein turnover via the ubiquitin-proteasome pathway. (a)** U2OS Flp-In T-REx cells stably expressing doxycycline (Dox)-inducible GFP-KLHL15-wt were cultivated in the absence ( − ) or presence ( + ) of Dox. Twenty-four hours post induction, cells were either mock-treated or treated with MG-132 (10 μM) for 6 h and whole-cell lysates were analysed by immunoblotting. **(b)** Twenty-four hours post induction, same cells as in **a** were either mock-treated or treated with MLN-4924 (100 nM) for 8 h and whole-cell lysates were analysed by immunoblotting. **(c)** Same cells as in **a** were transfected with either non-targeting (CNTL) or CUL3 siRNA oligos. 48 h later, GFP-KLHL15 expression was induced with Dox for 24 h and whole-cell lysates were analysed by immunoblotting. **(d)** U2OS cells were transfected with the indicated siRNA oligos for 48 h. Total cell extracts and chromatin-enriched fractions were analysed by immunoblotting using the indicated antibodies. **(e)** Same cells as in **a** were transfected with the indicated siRNA oligos. Twenty-four hours post transfection, cells were grown for 24 h in the absence ( − ) or presence ( + ) of Dox and whole-cell lysates were analysed by immunoblotting. **(f)** HEK293 Flp-In T-REx cells inducibly expressing GFP-CtIP were transfected with the indicated siRNAs. Forty-eight hours later, cells were transfected with His-Ub and the expression of CtIP was simultaneously induced with Dox. Eight hours post induction, cells were transfected with siRNAs for a second time. Three days after the first siRNA transfection, cells were treated with MG-132 (20 μM) for 6 h followed by lysis in buffer containing guanidium-HCl. Ubiquitin conjugates were pulled-down (PD) with Ni-NTA-agarose beads, eluted and analysed by immunoblotting with anti-GFP antibody. **(g)** Recombinant FLAG-CtIP was incubated with ATP, ubiquitin, E1, E2 enzymes and CUL3-N8/RBX1-KLHL15 (E3) for 30 min at 37 °C. Unmodified and ubiquitinated CtIP protein species were detected by immunoblotting with anti-FLAG antibody. Asterisks indicate neddylated CUL3.

polyubiquitinated species of CtIP upon MG-132 treatment (Fig. 2f). In contrast, however, CtIP ubiquitination was severely impaired in KLHL15- or CUL3-depleted cells (Fig. 2f). To prove that the CUL3-KLHL15 E3 ligase complex is sufficient to catalyse CtIP ubiquitination *in vitro*, we expressed and purified recombinant FLAG-CtIP and MBP-KLHL15 from HEK293T and Sf9 insect cells, respectively (Supplementary Fig. 2d). Recombinant CUL3/RBX1 was neddylated *in vitro* using the purified NEDD8 (N8) conjugation system[30]. Next, CtIP and KLHL15 were mixed with CUL3-N8/RBX1 complex and incubated with ATP, ubiquitin, E1 (UBE1) and E2 (Cdc34 or UbcH5b) enzymes. Western blot analysis revealed that CtIP was extensively modified when all components of the ubiquitination machinery were present in the reaction (Fig. 2g). Moreover, we found that this reaction was specifically catalysed by KLHL15 *in vitro*, since CtIP ubiquitination was not supported by KLHL21 (Supplementary Fig. 2e, lanes 11–13). As it is well-established that proteins targeted for proteasomal degradation are often conjugated to lysine 48 (K48) linked ubiquitin chains, we compared the efficiency of CtIP polyubiquitination by CUL3-KLHL15 between K48-only and K48R ubiquitin mutants using the same assay. We observed that CtIP ubiquitination was almost completely abrogated in presence of the K48R mutant, whereas chain formation was indistinguishable between wild-type and K48-only ubiquitin (Supplementary Fig. 2e). Collectively, our results demonstrated that CUL3-KLHL15 promotes K48-linked

CtIP polyubiquitination leading to its proteasome-dependent degradation.

**CtIP protein accumulates in KLHL15 knockout cells.** Since HEK293 cells were used in our proteomics screen for proteins that interact with CtIP, we rationalized that KLHL15 is adequately expressed in this cell line. Thus, we generated stable HEK293 KLHL15 knockout cells (HEK293$^{Cas9/KLHL15\Delta}$) using the CRISPR method introduced by Munoz et al.[31] We first screened several, single-cell clones for Cas9-induced disruption of KLHL15 protein by western blotting performed with a homemade rabbit polyclonal antibody raised against a recombinant fragment (amino acids 300–604) of human KLHL15 (Fig. 3a). Remarkably, and consistent with our previous findings, we observed a perfect positive correlation between the absence of KLHL15 and increase of CtIP protein levels (Fig. 3a). To further prove that the *KLHL15* gene was efficiently knocked out, the CRISPR-targeted genomic locus in HEK293$^{Cas9/KLHL15\Delta}$ clone 1 was PCR-amplified and subjected to deep-sequencing demonstrating that biallelic *KLHL15*-editing resulted in two truncated KLHL15 proteins harbouring only the N-terminal BTB domain (Supplementary Fig. 3a,b). Consistent with our previous data in U2OS cells (Fig. 2d), KLHL15 is

strongly associated with the chromatin compartment in HEK293 cells, further indicating that KLHL15 targets CtIP for ubiquitination and degradation in the nucleus (Fig. 3b). Notably, we found that KLHL15 loss does not affect cell cycle progression and, vice versa, that KLHL15 is expressed throughout the cell cycle (Supplementary Fig. 3c,d). Finally, we observed that CtIP protein half-life was dramatically prolonged in KLHL15 knockout cells and that MG-132 treatment did not alter CtIP protein stability in those cells, further supporting the importance of KLHL15 in governing CtIP protein turnover (Fig. 3c).

**KLHL15 expression sensitizes cells to CPT by inhibiting resection.** To investigate the role of KLHL15 in the DDR, we generated U2OS clones inducibly expressing GFP-tagged KLHL15-G386C and -Y552A, two mutants defective in CtIP binding (see Fig. 1f). However, in contrast to KLHL15-wt and KLHL15-Y552A, which localized predominantly in the nucleus and were present in the chromatin-enriched fraction, KLHL15-G386C was exclusively cytoplasmic and therefore excluded from further studies (Supplementary Fig. 4a,b). Consistent with our previous findings, CtIP protein levels were reduced in KLHL15-wt compared with KLHL15-Y552A mutant cells already at 24 h following induction of KLHL15 expression (Supplementary Fig. 4b).

In S and G2 phases of the cell cycle, CtIP is required for the processing of DSBs into ssDNA, triggering HR. Consequently, CtIP-deficient cells display reduced DNA-end resection and hypersensitivity towards CPT, a DNA topoisomerase inhibitor causing DSBs specifically at the replication fork[8]. In large agreement with that, we found that cells overexpressing KLHL15-wt, therefore harbouring very little amounts of CtIP, were hypersensitive to CPT (Fig. 4a). In response to DSBs, CtIP is required for RPA2 hyperphosphorylation, a commonly accepted marker for DNA-end resection[32]. Strikingly, the level of phosphorylated RPA2 upon CPT treatment was significantly reduced in KLHL15-wt compared with KLHL15-Y552A mutant cells (Fig. 4b and Supplementary Fig. 4c). Next, using a recently established flow-cytometry-based method to detect RPA retention on damaged chromatin[33], we observed that the percentage of CPT-induced RPA-positive cells was specifically reduced upon induction of KLHL15-wt expression (Fig. 4c), indicative of DNA-end resection defect. To directly measure the formation of ssDNA in response to CPT treatment, we employed the same approach but instead immunostained bromodeoxyuridine (BrdU)-labelled cells with an anti-BrdU antibody. Consistently, the BrdU signal was dramatically reduced in cells overexpressing KLHL15-wt, but not in the Y552A mutant cells (Fig. 4c). Importantly, the reported cellular phenotypes caused by KLHL15-wt overexpression were generally milder but followed the same trend as those caused by CtIP downregulation (Supplementary Fig. 4d–f). On the basis of these findings, we analysed the effect of KLHL15 expression on the efficiency of DSB repair by HR in established reporter cells[34]. As expected, transient transfection of these cells with KLHL15-wt, but not with KLHL15-Y552A, resulted in a significant reduction in HR frequency compared with control cells (Fig. 4d). Taken together, our data suggest that KLHL15 overexpression is detrimental to DNA-end resection and HR, most likely caused by enhanced CtIP proteasomal degradation.

**The C-terminal domain of CtIP mediates binding to KLHL15.** To gain further insights into the mechanism by which KLHL15 promotes CtIP protein turnover, we sought to identify the region within CtIP responsible for KLHL15 interaction. Human CtIP has two recognizable domains (Fig. 5a), a coiled-coil domain[35] and a highly conserved Sae2/Ctp1-like CTD, which is required for

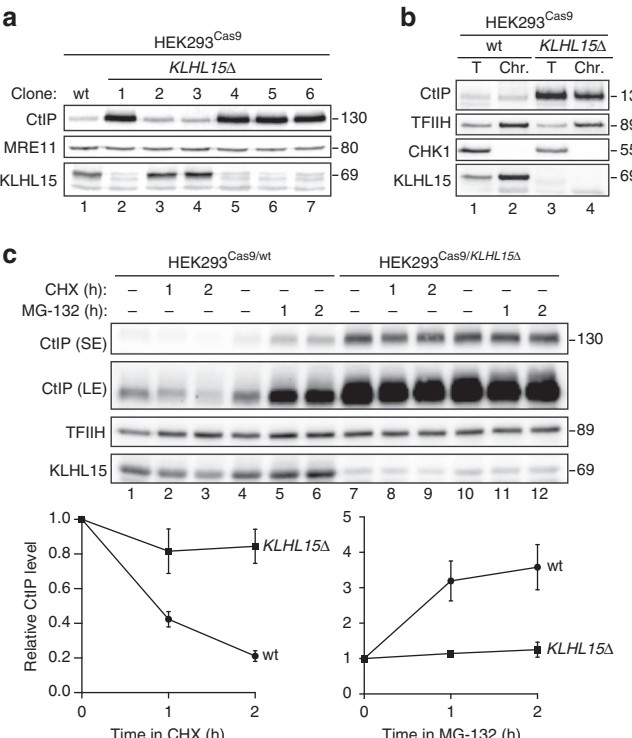

**Figure 3 | CtIP protein turnover is impaired in KLHL15 knockout cells.** (**a**) Lysates from a stable HEK293$^{Cas9}$ wild-type (wt) cell clone or from six different HEK293$^{Cas9}$ cell clones generated with the same sgRNA targeting KLHL15 (see Supplementary Fig. 3a) were subjected to western blotting with the indicated antibodies. (**b**) Total (T) cell extracts and chromatin-enriched (Chr.) fractions of HEK293$^{Cas9/wt}$ and HEK293$^{Cas9/KLHL15\Delta}$ cells were analysed by immunoblotting using the indicated antibodies. (**c**) Same cells as in **b** were either mock-treated, treated with CHX (100 µg ml$^{-1}$) or MG-132 (20 µM) for the indicated time points and lysates were subjected to western blotting with the indicated antibodies (upper panel). SE and LE; short- and long-exposure times of the same immunoblot. Relative CtIP protein levels were determined by quantification of CtIP band intensity (normalized to TFIIH) with the ImageJ software (lower panels). Data are represented as mean values of densitometric quantification ± s.e.m. ($n = 3$).

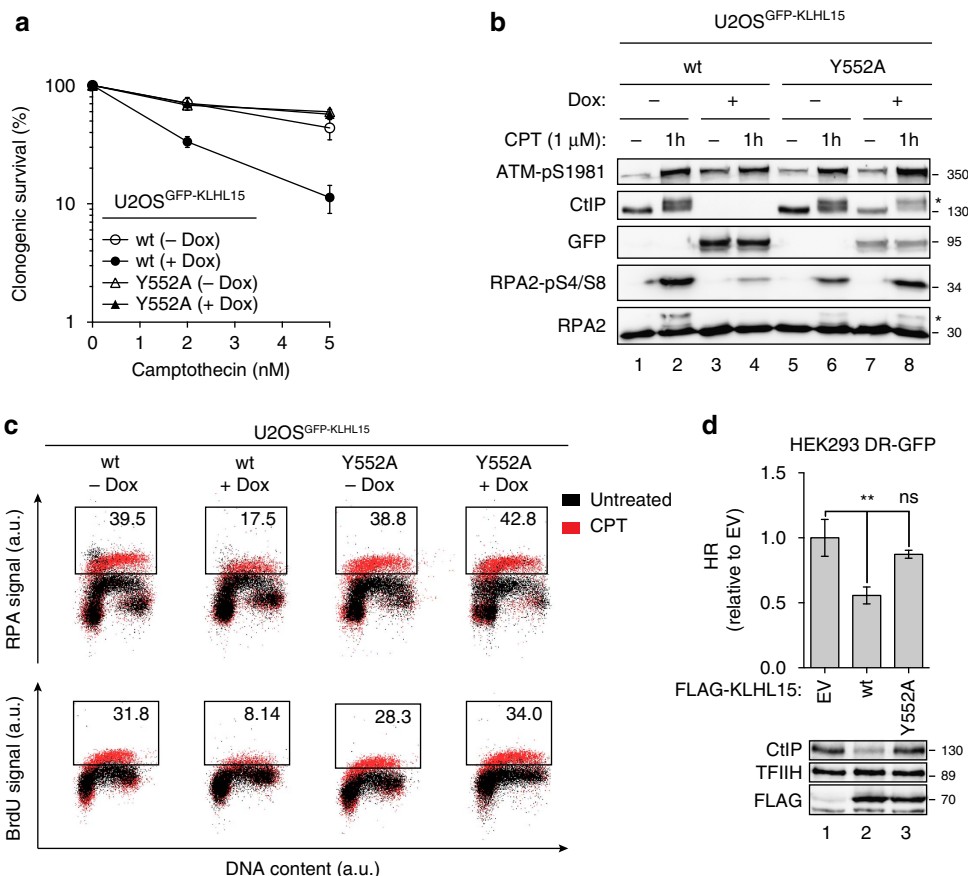

**Figure 4 | KLHL15 overexpression leads to camptothecin hypersensitivity and defective DNA-end resection.** (**a**) U2OS Flp-In T-REx cells inducibly expressing GFP-KLHL15-wt or GFP-KLHL15-Y552A were cultivated in the presence or absence of Dox. Twenty-four hours post induction, cells were treated with the indicated doses of camptothecin and survival was determined after 10 days by colony-formation assay. Data are presented as the mean ± s.d. ($n = 3$). (**b**) Same cells as in **a** were mock-treated or treated with camptothecin (CPT, 1 μM) for 1 h and lysates were analysed by immunoblotting using the indicated antibodies. Asterisks indicate hyperphosphorylated forms of CtIP and RPA2, respectively. (**c**) Same cells as in **b** were labelled with BrdU (30 μM) for 24 h before CPT treatment. Cells were harvested, permeabilized, fixed, immunostained with anti-RPA2 or anti-BrdU antibody and analysed by FACS. Dot plots representing the intensity of the signals for RPA2 or BrdU staining (y axis) against the DNA content (x axis). Quantification gates were established in untreated samples and the percentage of cells within the gates is indicated. (**d**) HEK293 DR-GFP cells were transfected with the *I-SceI* in combination with the indicated FLAG-KLHL15 expression plasmids and harvested after 48 h for flow cytometry and immunoblot analysis. Data are represented as mean ± s.d. ($n = 3$). Statistical analysis were carried out using unpaired, two-tailed *t*-tests. P values expressed as ** ($P < 0.005$) were considered significant. A.u., arbitrary units; FACS, fluorescence-activated cell sorting.

DNA-end resection[8]. In addition, CtIP contains several short sequence motifs (Fig. 5a) that are important for CtIP tetramerization[36–38] or for physical interactions with other proteins, such as FANCD2 (ref. 39), PIN1 (ref. 28), BRCA1 (refs 40,41) and CDH1 (ref. 42). Interestingly, by performing MBP-KLHL15 pull-down experiments from HEK293T cell lysates expressing different CtIP constructs, we found that GFP-CtIP-wt and GFP-CtIP-ΔN (deleted of amino acids (aa) 153–322) interacted with KLHL15, whereas GFP-CtIP-ΔC1 lacking the entire CTD (aa 790–897) did not (Fig. 5b). Moreover, when the same constructs were cotransfected with FLAG-KLHL15 into HEK293T cells, quantification of protein levels revealed that CtIP-ΔN showed rather variable abundance, whereas CtIP-ΔC1 was resistant to KLHL15 overexpression (Supplementary Fig. 5a). In fact, increased protein stability of a C-terminally truncated form of CtIP has been reported previously[43]. Consistently, we observed that ubiquitination of CtIP-ΔC1 *in vivo* (Fig. 5c) and *in vitro* (Supplementary Fig. 5b) was decreased compared with CtIP-wt. These results indicated that the CTD in CtIP is required for KLHL15 binding and subsequent ubiquitin-dependent proteolysis of CtIP.

To narrow down our search for a putative KLHL15-interaction motif in CtIP, we generated two additional CtIP C-terminal truncation mutants. Interestingly, GFP-CtIP-ΔC3 (aa 1–863) was still efficiently pulled-down and degraded by KLHL15, whereas GFP-CtIP-ΔC2 (aa 1–812) was not, suggesting that KLHL15 binds to a small region in the CTD of CtIP encompassing residues 813–863 (Fig. 5d and Supplementary Fig. 5c). In addition, by using pull-down assays, we found that a synthetic 15-mer peptide consisting of CtIP residues 830–844 (Fig. 5a) strongly interacted with GFP-KLHL15 expressed in cells as well as with endogenous KLHL15 in pull-down assay (Fig. 5e,f). Remarkably, the short peptide was effective in outcompeting the binding of endogenous CtIP to MBP-KLHL15, indicating that its amino acid sequence contained key residues required for KLHL15 interaction (Fig. 5g).

**Tyrosine residue 842 of CtIP is essential for KLHL15 binding.** To date, the protein phosphatase 2A (PP2A) regulatory subunit B′-beta is the only described substrate for the CUL3-KLHL15 E3 ubiquitin ligase[44]. This study established a tyrosine residue within a conserved stretch of three amino acids (FRY) as being critically

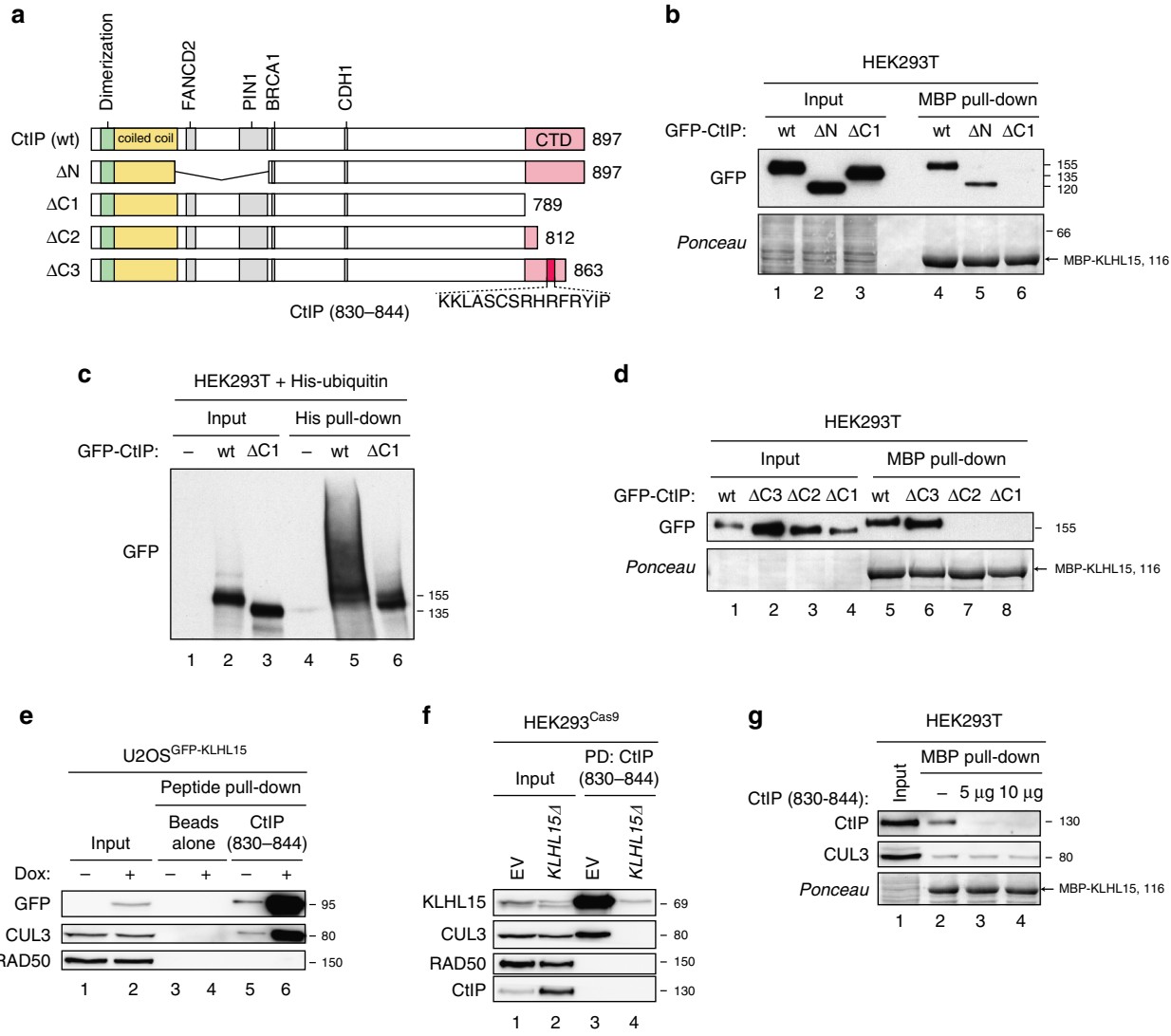

**Figure 5 | Mapping of a KLHL15-interacting region in CtIP.** (**a**) Schematic representation of GFP-tagged CtIP wild-type (wt) and truncation mutants. A 15-mer peptide encompassing amino acid residues 830–844 in the CtIP CTD important for KLHL15 binding is marked by a red box. (**b**) MBP-KLHL15 was coupled to amylose beads and incubated with lysates of HEK293T cells transfected with the indicated GFP-CtIP expression constructs for 48 h. Inputs and pulled-down protein complexes were analysed by immunoblotting. (**c**) HEK293T cells were cotransfected with the indicated GFP-CtIP expression constructs and His-Ubiquitin. Twenty-four hours post transfection cells were treated with MG-132 (20 μM) for 4 h. Cells were then lysed in buffer containing guanidium-HCl and ubiquitin conjugates were pulled-down using Ni-NTA-agarose beads, eluted, and analysed by immunoblotting with anti-GFP antibody. (**d**) MBP-KLHL15 pull-down assays were performed as in **b**. (**e**) Three microgram of a biotinylated 15-mer CtIP peptide (aa 830–844) indicated in **a** was coupled to streptavidin beads and incubated with lysates of U2OS^GFP-KLHL15-wt cells cultivated in absence (−) or presence (+) of Dox for 24 h. Inputs and pulled-down protein complexes were analysed by immunoblotting. (**f**) Three microgram of the CtIP peptide coupled to streptavidin beads was incubated with lysates of HEK293^Cas9/wt and HEK293^Cas9/KLHL15Δ cells. Inputs and pulled-down protein complexes were analysed by immunoblotting. (**g**) Recombinant MBP-KLHL15 was coupled to amylose beads and incubated with lysates of HEK293T cells in the absence (−) or presence of the indicated amounts of the CtIP 15-mer peptide. Inputs and pulled-down protein complexes were analysed by immunoblotting.

important for KLHL15-mediated degradation of PP2A/B'-beta. Strikingly, when we looked for the presence of a similar motif in human CtIP, sequence alignments highlighted a well-conserved FRY motif (aa 840–842) located exactly within the region previously shown to mediate KLHL15 interaction (Fig. 6a). Of note, CtIP homologues in *Caenorhabditis elegans*, *Dictyostelium discoideum* and *Schizosaccharomyces pombe* did not contain a FRY motif at this position (Fig. 6a), which could be partly explained by the fact that BTB-BACK-Kelch family proteins are extremely rare in worms and completely absent in slime molds and yeast[45].

To investigate whether the CtIP FRY motif is involved in KLHL15 recognition, we first generated a triple-alanine

substitution mutant (GFP-CtIP-AAA) and observed that the protein was partially resistant to KLHL15-mediated degradation (Supplementary Fig. 6a). Next, we focused on elucidating the role of Y842 in being the prime determinant for the physical interaction between CtIP and KLHL15, as shown for Y52 in PP2A/B'-beta[44]. Indeed, pull-down assays using bacterially expressed GST-CtIP-CTD as bait showed that the replacement of tyrosine 842 with alanine (Y842A) severely impaired binding of the CUL3-KLHL15 complex (Fig. 6b), whereas the previously established interaction with MRE11 (refs 8,46,47) remained largely unaffected (Supplementary Fig. 6b,c). Next, we subjected cell lysates expressing GFP-CtIP to IP using GFP antibody beads and found that the Y842A mutant exhibited reduced binding to

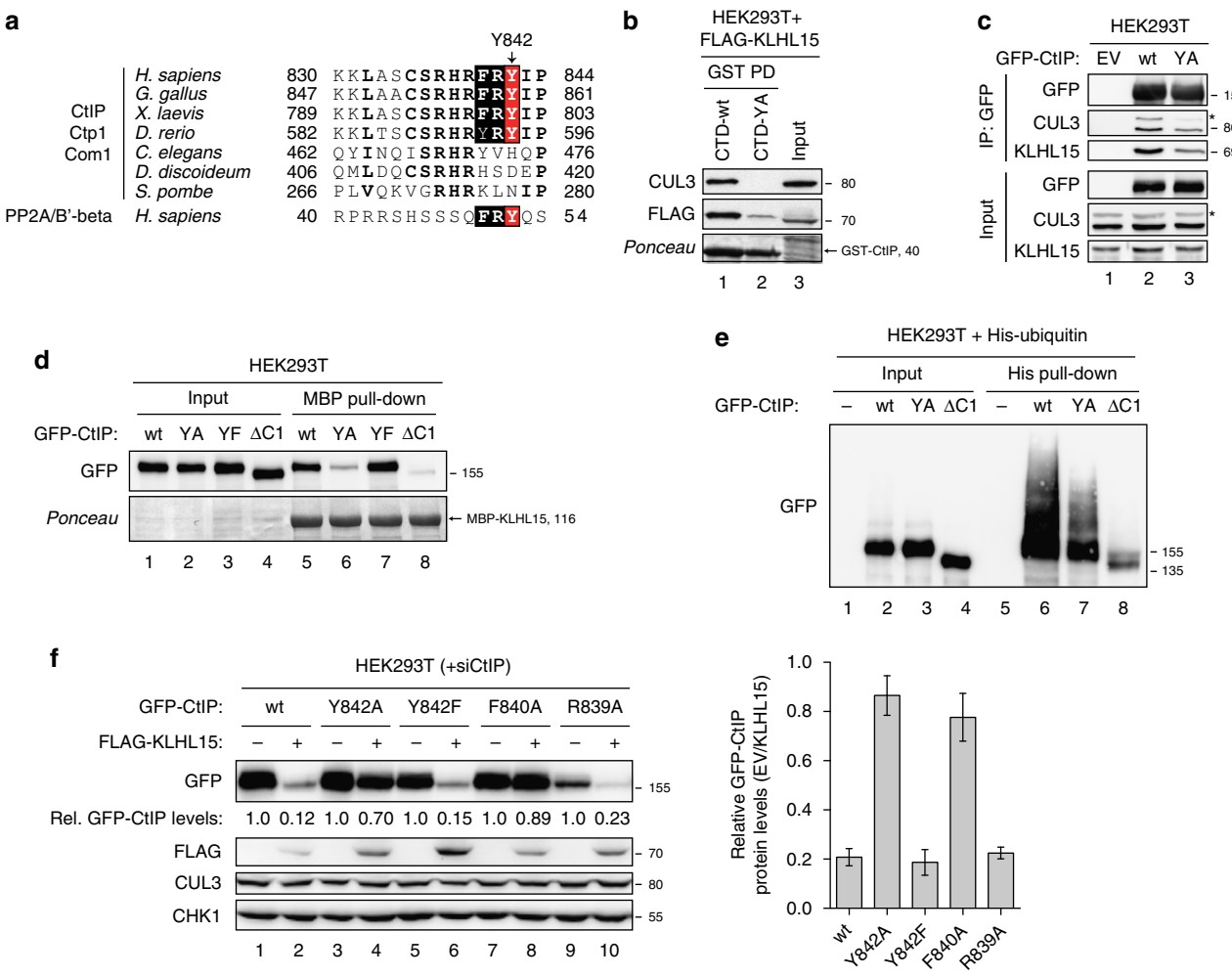

**Figure 6 | Identification of a conserved FRY motif in vertebrate CtIP essential for KLHL15 interaction.** (**a**) Alignment of the KLHL15-interacting region in human CtIP (aa 830–844) with the corresponding region in CtIP/Ctp1/Com1 orthologs of other species. A part of the KLHL15 binding domain in human PP2A/B'-beta is shown below[44]. The rectangular box represents the putative KLHL15-binding motif (FRY). The highly conserved tyrosine residue (Y842 in human CtIP) is highlighted in red. Other, highly conserved amino acid residues are marked in bold typeface. (**b**) Bacterially expressed GST fusion proteins of the CtIP CTD (aa 790–897), either wt or Y842A mutant, were coupled to glutathione sepharose beads and incubated with lysates from HEK293T cells transfected with the FLAG-KLHL15 expression plasmid for 48 h. Pulled-down protein complexes and the input were analysed by immunoblotting. (**c**) HEK293T cells were transfected with either empty vector (EV) or the GFP-CtIP expression constructs. 48 h after transfection, cells were lysed and whole-cell extracts were subjected to IP using anti-GFP affinity resin. Inputs and recovered protein complexes were analysed by immunoblotting. (**d**) Recombinant MBP-KLHL15 was coupled to amylose beads and incubated with lysates from HEK293T cells transfected with the indicated GFP-CtIP expression constructs for 48 h. Inputs and pulled-down protein complexes were analysed by immunoblotting. (**e**) HEK293T cells were cotransfected with the indicated GFP-CtIP constructs and His-Ubiquitin. Twenty-four hours post transfection cells were treated with MG-132 (20 µM) for 4 h. Cells were then lysed in buffer containing guanidium-HCl and ubiquitin conjugates were pulled-down using Ni-NTA-agarose beads, eluted and analysed by immunoblotting with anti-GFP antibody. (**f**) HEK293T cells were transfected with CtIP siRNA and 24 h later cotransfected with the indicated siRNA-resistant GFP-CtIP expression constructs and FLAG-KLHL15. Forty-eight hours post siRNA transfection cells were analysed by immunoblotting (left). The GFP-CtIP signal intensities were quantified using ImageJ and represented as EV/FLAG-KLHL15 ratios (right). Data are represented as mean values of densitometric quantification ± s.e.m. ($n \geq 3$). Asterisks indicate neddylated CUL3.

endogenous KLHL15 and CUL3 as compared with wild-type protein (Fig. 6c). Likewise, MBP-pull-down assays showed decreased interaction between KLHL15 and CtIP-Y842A (Fig. 6d). Importantly, using the same approach, we found that replacing Y842 with a non-phosphorylatable phenylalanine completely restored KLHL15-CtIP interaction (Fig. 6d), indicating that Y842 phosphorylation is not required for KLHL15 binding, whereas the side-chain aromatic ring at this position is. As a functional consequence of reduced KLHL15 interaction, we observed that the CtIP-Y842A mutant was partially defective in polyubiquitination *in vivo* (Fig. 6e). Consistent with these findings, CtIP-Y842A was resistant to

KLHL15 overexpression, whereas CtIP-Y842F was degraded to the same extent as CtIP-wt (Fig. 6f). To examine whether the FRY motif indeed constitutes a canonical docking site for KLHL15, we constructed two additional CtIP mutants in which F840 and R839, located in the conserved neighbouring 'RHR' motif, were substituted with alanine residues (Fig. 6a). We again cotransfected the GFP-tagged versions together with FLAG-KLHL15 and found that F840A behaved identical to Y842A in terms of being resistant to KLHL15 overexpression, whereas R839A was degraded to a similar extent as compare to wild-type (Fig. 6f). Taken together, these findings indicate that the FRY motif and Y842 in particular are essential for KLHL15

recognition and subsequent degradation of CtIP by the ubiquitin-proteasome pathway.

**KLHL15 regulates DNA-end resection and DSB repair**. To investigate the role of CtIP ubiquitination by KLHL15 in response to DNA damage, we generated stable U2OS cell clones inducibly expressing siRNA-resistant GFP-tagged CtIP-wt or CtIP-Y842A. Importantly, cell cycle profiles, CtIP nuclear localization and accumulation to DSB-containing tracks generated by laser microirradiation were highly similar in both cell lines, indicating that these transgenes are functional (Supplementary Fig. 7a–c). Interestingly, cycloheximide chase experiments revealed that CtIP-Y842A had a much prolonged half-life compared with CtIP-wt, a phenotype reminiscent of the reduced CtIP protein turnover in KLHL15 knockout cells (Supplementary Fig. 7d). However, CtIP-Y842A efficiently rescued CPT hypersensitivity of CtIP-depleted cells, suggesting that increased CtIP protein stability does not negatively affect CtIP function in DSB processing (Supplementary Fig. 7e). Next, we compared CPT-induced ATM/ATR activation between the two cell lines by western blotting to assess whether impairment of KLHL15 binding to CtIP has an impact on DNA damage signalling. Interestingly, we found that ATR-mediated CHK1 and RPA2 phosphorylation was increased in CtIP-Y842A mutant compared with CtIP-wt cells, indicative of elevated DNA-end resection, whereas ATM autophosphorylation remained unaltered (Fig. 7a). Moreover, Y842A rescued defective RPA2 hyperphosphorylation in CtIP-depleted cells to a much higher extent as compared with control cells (Fig. 7b). Next, we analysed RPA accumulation on damaged chromatin as well as the formation of ssDNA by flow-cytometry. Consistent with our immunoblot analysis, Y842A led to increased RPA chromatinization and ssDNA formation upon CPT treatment, further demonstrating that impaired CtIP protein turnover causes hyper-resection of DSBs (Fig. 7c). Underscoring the importance of KLHL15 in regulating DNA-end resection, we observed that KLHL15 knockout cells display elevated RPA2 phosphorylation levels as compared with control cells (Fig. 7d). Importantly, siRNA-mediated downregulation of CtIP in KLHL15 knockout cells suppressed the hyper-resection phenotype (Fig. 7d), indicating that KLHL15 limits resection by promoting CtIP proteasomal degradation and that CtIP is most likely the key substrate of KLHL15 involved in the DDR. Next, using the flow-cytometry-based assay to quantify DNA-end resection, we observed that the amount of RPA-bound ssDNA in KLHL15 knockout cells was far higher than in control cells (Fig. 7e). Furthermore, loss of KLHL15 resulted in a marked increase of RPA2 hyperphosphorylation after treatment with ionizing radiation (IR) (Fig. 7f). DNA-end resection is inhibitory to the repair of DSBs by NHEJ. With regards to this view and since NHEJ is the predominant repair mechanism for IR-induced two-ended DSBs, we next addressed the survival of KLHL15 knockout cells following IR treatment using clonogenic assay. Remarkably, HEK293^Cas9/KLHL15Δ cells were hypersensitive to IR, indicative of compromised NHEJ activity (Fig. 7g). To investigate whether regulation of CtIP protein turnover by KLHL15 plays a direct role in DSB repair pathway choice, we measured NHEJ or HR frequencies in HEK293 GFP-reporter cells[34]. First, we discovered that KLHL15 knockdown caused a significant reduction in NHEJ, similar to that seen after depletion of the canonical NHEJ factor XRCC4 (Fig. 7h). In large agreement with this finding, NHEJ frequency was decreased upon overexpression the CtIP-Y842A mutant, further supporting the concept that excessive DNA-end resection is counterproductive for NHEJ (Supplementary Fig. 7f). Next, we performed HR reporter assays and observed that downregulation of KLHL15 coincided with

increased HR efficiency (Fig. 7i), whereas CtIP-Y842A had no major impact on homology-directed repair of DSBs (Supplementary Fig. 7g). Altogether, this data provide evidence that KLHL15 is a key factor governing DNA-end resection and DSB repair pathway choice through regulating CtIP ubiquitination and, ultimately, CtIP protein turnover.

**PIN1 and KLHL15 cooperate in promoting CtIP degradation**. In an earlier study, we have reported that PIN1, a phosphorylation-specific prolyl isomerase, promotes CtIP degradation by the ubiquitin-proteasome pathway[28]. However, the responsible E3 ligase working in concert with PIN1 in CtIP ubiquitination has not been identified yet. Specifically, we demonstrated that disruption of this regulatory function by PIN1 depletion or expression of a CtIP mutant refractory to PIN1 binding resulted in hyper-resection and reduced NHEJ, meanwhile PIN1 overexpression suppressed resection and HR. As these cellular phenotypes are highly reminiscent of those observed in this study, we speculated whether CtIP isomerization by PIN1 may facilitate KLHL15-dependent CtIP degradation. Interestingly, we found that CtIP S276A/T315A (2A) impaired in PIN1 binding[28] was partially resistant to KLHL15 overexpression (Fig. 8a). In contrast, the CtIP KEN-box mutant (K467A) defective in CDH1 binding and subsequent polyubiquitination by the APC/C^Cdh1 E3 ligase[29], was as efficiently degraded by KLHL15 as the wild-type CtIP protein (Fig. 8a). This result indicated a possible functional connection between PIN1 and KLHL15 in controlling CtIP protein turnover. To address this further, we next investigated whether CtIP isomerization by PIN1 may foster CtIP-KLHL15 interaction. MBP-pull-down assays revealed that the GFP-CtIP-2A mutant protein was fully proficient in binding to recombinant KLHL15 purified from insect cells (Supplementary Fig. 8a). Interestingly, however, KLHL15-CtIP interaction was reduced in cells depleted for PIN1 (Fig. 8b), indicating that isomerization of CtIP is not essential for physical interaction between KLHL15 and CtIP, yet that it somehow aids the formation of a ternary CUL3-KLHL15-CtIP complex in cells. Consistent with this hypothesis, simultaneous downregulation of KLHL15 and PIN1 did not further reduce CtIP ubiquitination when compared with the single knockdowns (Supplementary Fig. 8b). As reported previously[28], PIN1 overexpression destabilized CtIP particularly in response to DSBs causing reduced RPA2 hyperphosphorylation (Fig. 8c, lanes 1–4). We now observed that both phenotypes can be reversed by transfecting PIN1 into KLHL15 knockout cells, indicating that PIN1-mediated CtIP degradation after DNA damage requires KLHL15 (Fig. 8c, lanes 5–8). Taken together, these experiments suggested that the PIN1 isomerase, at least in certain scenarios, facilitates CtIP ubiquitination and proteasomal degradation through the CUL3-KLHL15 E3 ligase.

**Discussion**
Ubiquitination and its crosstalk with other post-translational modifications have recently emerged as key regulatory mechanisms in the cellular response to DNA damage[48,49]. As a consequence, defects in the ubiquitination machinery have been shown to be associated with reduced cellular survival and increased genomic instability in response to DSBs, mainly due to impaired protein–protein interactions causing mislocalization of DSB repair factors[50–52]. Of note, these processes mostly involve non-proteasomal ubiquitin signalling instead of the classical function of E3 ubiquitin ligases in targeting substrates to proteasome-dependent degradation.

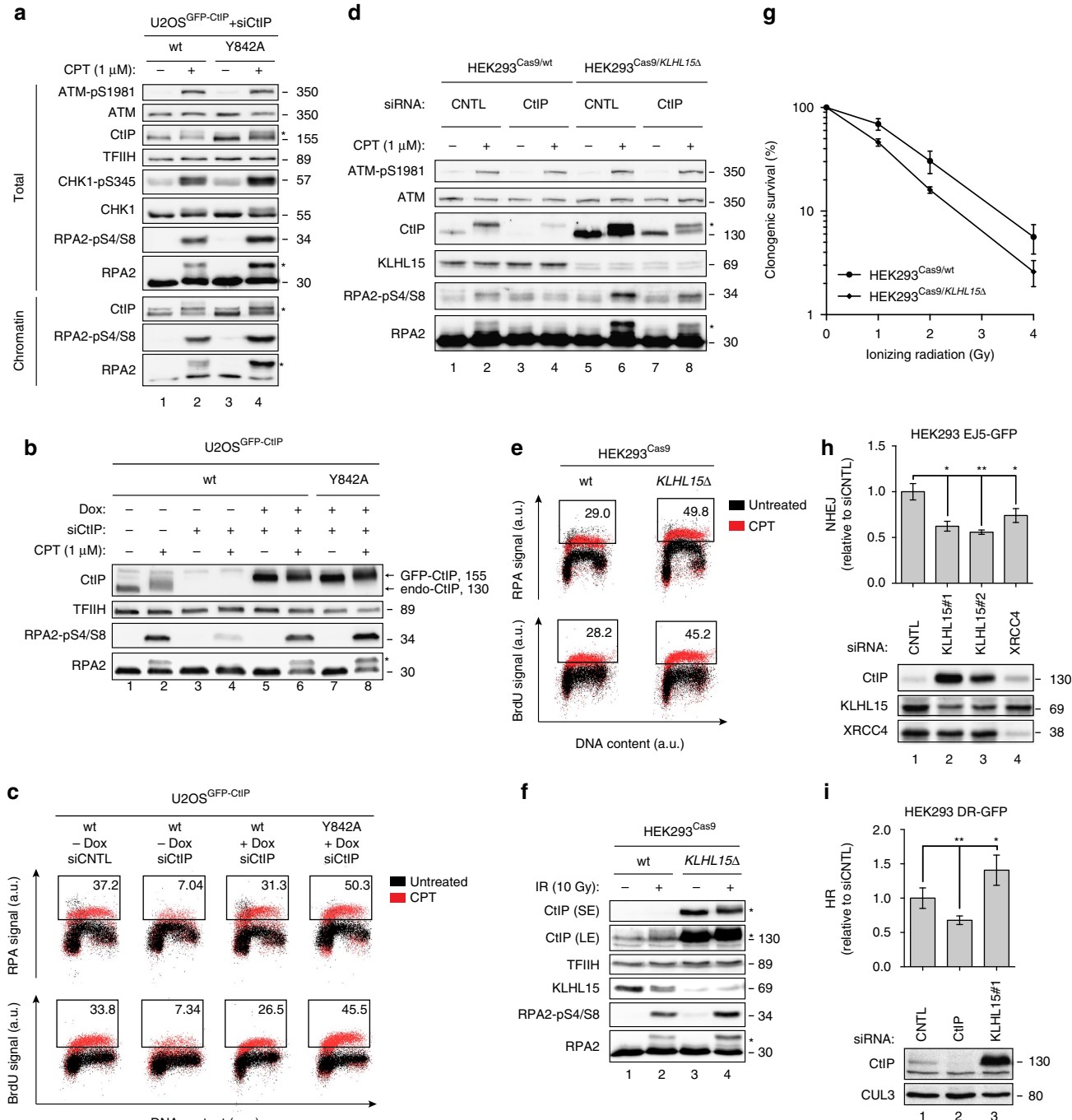

**Figure 7 | KLHL15-dependent CtIP ubiquitination governs DNA-end resection and DSB repair pathway choice.** (**a**) U2OS cells stably expressing doxycycline (Dox)-inducible siRNA-resistant GFP-CtIP-wt or GFP-CtIP-Y842A were transfected with CtIP siRNA. 24 h later, GFP-CtIP expression was induced with Dox. Forty-eight hours post siRNA-transfection, cells were either mock-treated or treated with CPT (1 μM) for 1 h. Total cell extracts and chromatin-enriched fractions were analysed by immunoblotting. (**b**) Same cells as in **a** were transfected with the indicated siRNA oligos. One day later, GFP-CtIP expression was induced for 24 h and cells were treated with CPT for 1 h before western blot analysis. (**c**) Same cells as in **b** were simultaneously labelled with BrdU (30 μM). 48 h post siRNA-transfection, cells were treated with CPT for 1 h, immunostained with anti-RPA2 or anti-BrdU antibody and analysed by FACS. Dot plots represent the intensity of the signals for RPA2 or BrdU staining (y axis) against the DNA content (x axis). Quantification gates were established in untreated samples and the percentage of cells within the gates are indicated. (**d**) HEK293$^{Cas9/wt}$ and HEK293$^{Cas9/KLHL15\Delta}$ cells were transfected with the indicated siRNA oligos. Forty-eight hours later, cells were treated with CPT for 1 h and analysed by immunoblotting. (**e**) Same cells as in **d** were processed for FACS analysis as in **c**. (**f**) Same cells as in **d** were irradiated at 10 Gy and 3 h later analysed by immunoblotting. SE and LE denote short and long exposures of the same membrane. (**g**) Same cells as in **d** were irradiated with increasing doses of IR and survival was determined by colony-formation assay. Data are presented as the mean ± s.d. ($n = 4$). (**h**,**i**) HEK293 EJ5-GFP or DR-GFP cells were transfected with the indicated siRNAs. Two days later, cells were transfected with the I-SceI expression plasmid and harvested after 48 h for flow cytometry and immunoblot analysis. Data in **h** and **i** are represented as mean ± s.e.m. ($n = 3$) and as mean ± s.d. ($n = 4$), respectively. Unpaired, two-tailed t-tests were performed and P values expressed as *($P < 0.05$) and **($P < 0.005$) were considered significant. Asterisks indicate hyperphosphorylated forms of CtIP and RPA2. A.u., arbitrary units; FACS, fluorescence-activated cell sorting.

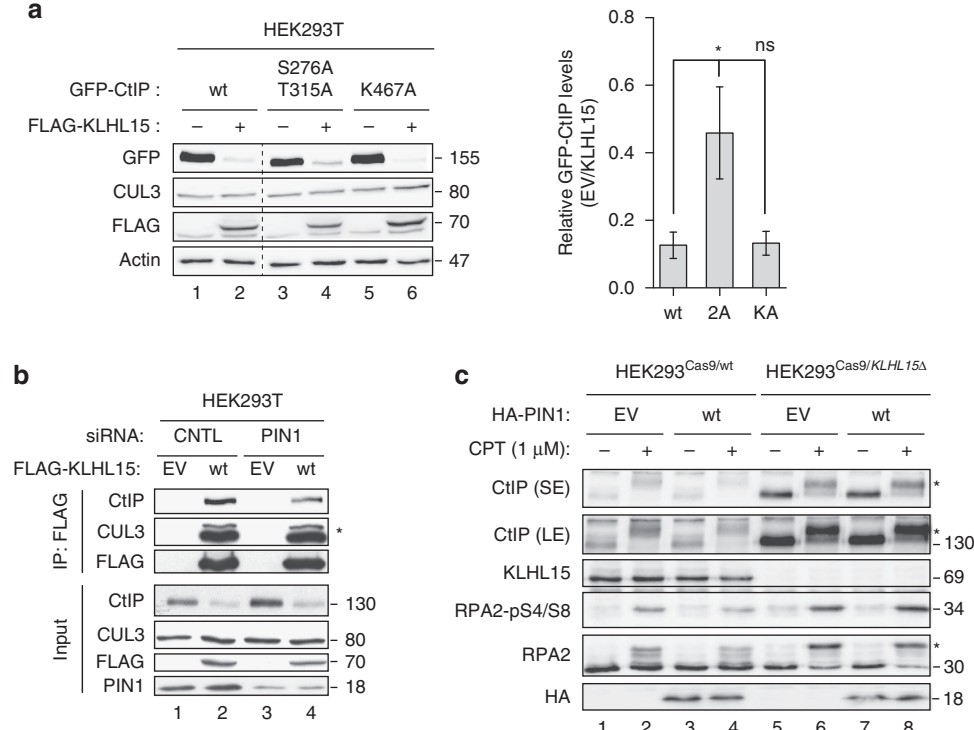

**Figure 8 | PIN1 isomerase facilitates KLHL15-dependent degradation of CtIP.** (**a**) HEK293T cells were cotransfected with FLAG-KLHL15 and either GFP-CtIP-wt, -S276A/T315A (2A, defective in PIN1 interaction), or -K467A (KA, defective in Cdh1 interaction) expression constructs. Forty-eight hours post transfection, cells were analysed by immunoblotting (left). Relative CtIP protein levels were determined by quantification of CtIP band intensity (normalized to Actin) with the ImageJ software (right). Data are represented as mean values of densitometric quantification ± s.e.m. ($n \geq 3$). Statistical analysis was carried out using unpaired, two-tailed $t$-tests. $P$ values expressed as *($P < 0.05$) was considered significant. (**b**) HEK293T cells were transfected twice with the indicated siRNA oligos for two consecutive days. Forty-eight hours after the first siRNA transfection, cells were transfected with either empty vector (EV) or FLAG-KLHL15 expression constructs. 72 h after the first siRNA transfection, cells were lysed and whole-cell extracts were subjected to IP using anti-FLAG M2 affinity resin. Inputs and recovered protein complexes were analysed by immunoblotting. The asterisk indicates neddylated CUL3. (**c**) HEK293$^{Cas9/wt}$ and HEK293$^{Cas9/KLHL15\Delta}$ cells were transfected with EV or HA-PIN1 expression construct. Forty-eight hours after transfection, cells were either mock-treated or treated with CPT (1 µM) for 1 h. Whole-cell lysates were analysed by immunoblotting using the indicated antibodies. SE and LE denote short and long exposures of the same membrane. Asterisks indicate hyperphosphorylated forms of CtIP and RPA2, respectively.

Here, we demonstrate that proteasomal degradation triggered by a CRL3 E3 ligase plays a prominent role in the regulation of DSB repair. Specifically, our results indicate that the BTB-BACK-Kelch protein KLHL15 acts as an adaptor for CUL3 to target the DNA-end resection factor CtIP for ubiquitination and subsequent proteasomal degradation (Fig. 9). We further define the molecular basis of the CUL3-KLHL15-CtIP ternary complex by showing that KLHL15 interacts with CUL3 *via* its canonical '3-box' motif and recruits CtIP to CUL3 via its C-terminal Kelch domain. Moreover, our data reveals that the FRY tripeptide sequence, present in the C-terminus of CtIP, may serve as general KLHL15 docking motifs present in substrate proteins targeted by the CUL3-KLHL15 E3 ubiquitin ligase. Specifically, we report that CtIP-Y842, located within a conserved 'FRY motif', is essential for KLHL15-mediated CtIP ubiquitination and subsequent degradation.

CtIP is required for DNA-end resection, the initial step of homology-directed repair, and is therefore a key factor in controlling repair pathway choice as processed DSB ends can no longer serve as substrates for NHEJ[53]. Collectively, our findings support a model in which the CUL3-KLHL15 ubiquitin ligase governs the turnover of CtIP to fine-tune the balance between HR and NHEJ (Fig. 9). Accordingly, cells that express high levels of KLHL15 exhibit unscheduled and enhanced CtIP degradation, ultimately leading to strong defects in DNA-end

resection and increased channelling of DSB repair from error-free HR into error-prone NHEJ (Fig. 9). In contrast, cells either lacking KLHL15 or expressing a mutant of CtIP refractory to KLHL15 binding display aberrant accumulation of CtIP and excessive DNA-end resection, thereby suppressing DSB repair by NHEJ (Fig. 9). Recently, a similar regulatory function in DSB repair has been proposed for Keap1, another CUL3 substrate adaptor. However, rather than involving the proteasome, the authors of this study showed that Keap1-dependent ubiquitination of PALB2 suppresses its interaction with BRCA1, thereby prohibiting HR in G1 cells[54].

Previously, we reported that PIN1 isomerase tips the balance between HR and NHEJ by a very similar mechanism involving CtIP ubiquitination and degradation[28]. Here, we provide circumstantial evidence that CUL3-KLHL15 is the responsible E3 ligase operating in conjunction with PIN1 to control CtIP protein turnover. Interestingly, we find that isomerization of CtIP is not essential for the direct physical interaction between CtIP and KLHL15, but that PIN1 is important for stable complex formation of CUL3-KLHL15-CtIP, thereby reinforcing CtIP polyubiquitination and degradation in specific cellular contexts. For instance, as PIN1 binding to CtIP requires initial phosphorylation of CtIP by proline-directed kinases (for example, cyclin-dependent kinases), we previously proposed that PIN1 triggers CtIP ubiquitination and degradation predominantly

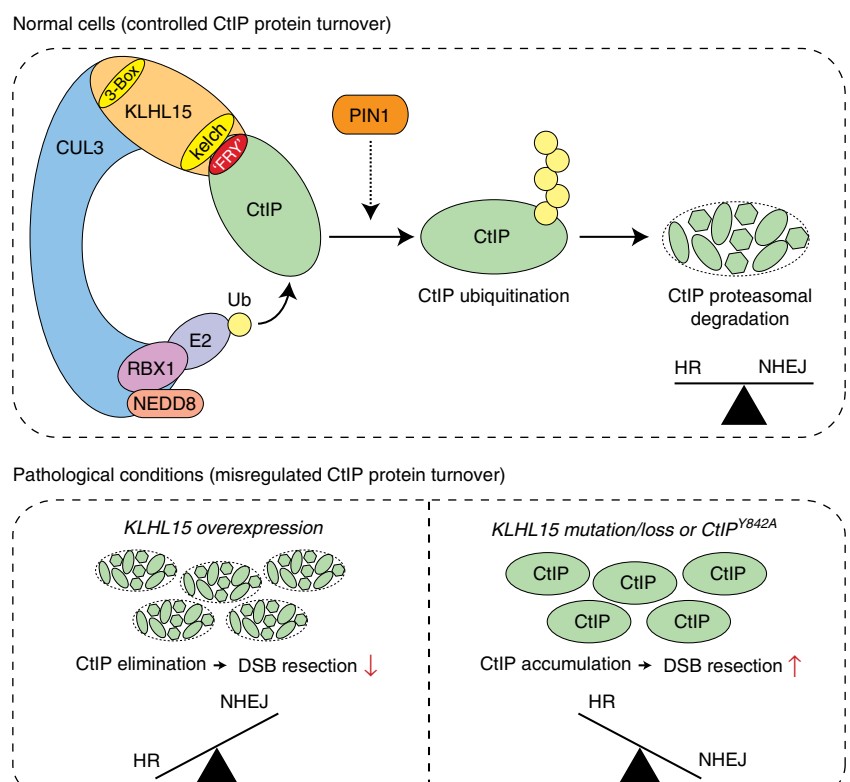

**Figure 9 | CtIP ubiquitination by CUL3-KLHL15 promotes CtIP proteasomal degradation to fine-tune DNA-end resection and DSB repair pathway choice.** Schematic model depicting the molecular architecture of the CUL3-KLHL15 E3 ubiquitin ligase complex and the functional consequences of KLHL15-mediated CtIP protein turnover in controlling the balance between NHEJ and HR in normal KLHL15-expressing cells (upper part) and under pathological conditions of altered KLHL15 expression (lower part). CtIP isomerization by PIN1 triggers CtIP proteasomal degradation[28] at least in part by facilitating CtIP ubiquitination by the CUL3-KLHL15 E3 ligase complex. See discussion for more details.

upon DNA damage in late S and $G_2$ phases of the cell cycle[28]. We now conclude that PIN1 and CUL3-KLHL15 collaborate in fine adjusting CtIP protein stability to attenuate DNA-end resection activity whenever NHEJ is the preferred mode of DSB repair. Interestingly, a similar mechanism has been described for PML degradation, which, in response to hypoxia, gets first isomerized by PIN1 and then ubiquitinated by the CUL3-KLHL20 E3 ubiquitin ligase[55]. Clearly, further investigations are required to elucidate how, at the molecular level, isomerization-induced conformational changes in CtIP stimulate its ubiquitination by CUL3-KLHL15 interaction.

Moreover, it also remains to be determined under which physiological conditions CUL3-KLHL15 primes CtIP for degradation in otherwise unperturbed cells. So far, studies on Keap1 and KLHL20 revealed a possible mode for regulating ubiquitination, which occurs at the level of the CUL3 adaptor rather than at the level of the substrate. CUL3-Keap1 constitutively targets Nrf2 until oxidative stress modifies several redox-sensitive cysteine residues of Keap1 interfering with the E3-ligase activity[56,57]. Similarly, DAPK is constitutively degraded by CUL3-KLHL20, until stress-induced interferon production causes sequestering of KLHL20 into PML nuclear bodies and thus away from its substrate[58]. In analogy to that, one could envision that KLHL15 controls the basal protein level of CtIP to keep DNA-end resection in check until certain stimuli modulate CUL3-KLHL15 ubiquitin ligase activity to either accelerate or delay CtIP protein turnover, thereby guiding DSB repair towards either NHEJ or HR. Furthermore, we observed that KLHL15 is predominantly chromatin-bound, suggesting that it may only target CtIP for degradation in genomic loci where error-prone

repair by NHEJ is not detrimental to the cell. On the other hand, KLHL15 may be absent from transcriptionally active chromatin to allow DNA-end resection and HR[59].

We also anticipate that KLHL15 function may be regulated at multiple levels, including gene expression and subcellular localization, as previously reported for KLHL20 (ref. 60). Along this line, KLHL15 mRNA levels were reported to be highest in lung, muscle and spleen, suggesting tissue-specific regulation of KLHL15 expression[44]. Interestingly, publicly available gene expression databases, such as MediSapiens, indicate highest KLHL15 expression levels predominantly in hematopoietic stem cells and in all four main types of leukemia (http://ist.medisapiens.com/#ENSG00000174010). Notably, the role for CtIP ubiquitination by KLHL15 in the regulation of DSB repair was most likely acquired later during evolution, as both KLHL15 as well as the KLHL15-binding motif in CtIP (FRY) are highly conserved in vertebrates but missing in yeast and worms.

Based on our data and the study of Oberg et al.[44] on KLHL15-dependent degradation of PP2A/B'-beta, we further hypothesize that the FRY tripeptide sequence may constitute a consensus KLHL15-Kelch domain interaction motif. In analogy to this, the CUL3 adaptor Keap1 recognizes its substrates including NRF2 and PALB2 via a conserved E(S/T)GE motif[61–63]. Moreover, we demonstrate that a CtIP-Y842F mutant is proficient in being targeted for degradation by KLHL15, excluding the possibility that Y842 phosphorylation is a prerequisite for KLHL15 binding. Nevertheless, phosphorylation of Y842 may negatively interfere with CtIP-KLHL15 interaction. Consistent with this hypothesis, inhibition of substrate recruitment to CRL3s by phosphorylation

has been previously reported[19]. For instance, NRF2 threonine phosphorylation of the aforementioned ETGE motif disrupts its interaction with Keap1 (ref. 26). It did also not escape our notice that the KLHL15-binding motif is in immediate proximity to an 'RHR' motif recently shown to be important for *S. pombe* Ctp1 binding to DNA *in vitro*[38]. However, based on the fact that the 'FRY motif' is not conserved in yeast and considering that, according to our data, CtIP-R839A is still degraded by KLHL15, we think it is reasonable to conclude that CtIP ubiquitination by KLHL15 (mediated through the 'FRY motif') and CtIP binding to DNA (mediated through the 'RHR motif') are mutually exclusive events in the regulation of DNA-end resection.

Lastly, our findings may have important therapeutic implications for some cancer types displaying KLHL15 overexpression. For example, high KLHL15 protein levels may render cancer cells hypersensitive to DNA topoisomerase inhibitors. Likewise, mutations of KLHL15 may lead to aberrant activation of DNA-end resection and HR-mediated DSB repair, which in turn could again provide the opportunity to design more effective personalized therapeutic strategies. In this respect, over 90 cancer-associated somatic mutations of KLHL15 are currently recorded in the Catalogue of Somatic Mutations in Cancer (COSMIC) database (www.cancer.sanger.ac.uk), but their molecular effects on cancer pathogenesis require further investigations.

## Methods

**Cell culture.** U2OS and HEK293T cells (Invitrogen, Life Technologies) were grown in DMEM supplemented with 10% FCS, 100 U ml$^{-1}$ penicillin, and 100 µg ml$^{-1}$ streptomycin. U2OS Flp-In T-REx (a kind gift from Daniel Durocher, University of Toronto) and HEK293 Flp-In T-Rex (Invitrogen, Life Technologies) cells were maintained in medium supplemented with 10 µg ml$^{-1}$ blasticidin and 300 µg ml$^{-1}$ zeocin. The Flp-In T-REx system (Invitrogen Life Technologies) was used to generate cell lines stably expressing different siRNA-resistant GFP-CtIP or GFP-KLHL15 constructs under the control of a doxycycline-inducible promoter. In brief, expression vectors pcDNA5/FRT/TO-GFP-CtIP or pcDNA5/FRT/TO-GFP-KLHL15 and the Flp recombinase expression plasmid, pOG44, were mixed in a 1:9 ratio and transfected into U2OS Flp-In T-REx cells (a kind gift of Daniel Durocher, University of Toronto) using FuGENE6 Trafection Reagent (Promega). 24 h later, cells were plated at different dilutions and 48 h post-transfection the medium was supplemented with 250 µg ml$^{-1}$ hygromycin B and 12.5 µg ml$^{-1}$ blasticidin S. The medium was replaced every 2–3 days and cells were selected for approximately 14 days. Resistant colonies were picked and single-cell clones analysed for GFP expression by immunoblotting and immunofluorescence microscopy. The CtIP-Y842A mutation in stable U2OS cells was further verified by genomic sequencing. HEK293$^{Cas9/wt}$ and HEK293$^{Cas9/KLHL15\Delta}$ cell lines were generated as described previously[31]. In brief, HEK293/Flp-In cells inducibly expressing Cas9-FLAG (HEK293$^{Cas9}$) were transfected with a plasmid (DU46129, U6_gRNA_empty vector) encoding a short guide RNA (sgRNA) targeting KLHL15 or the empty vector as control. Eight hours after transfection Cas9-FLAG expression was induced with Dox (1 µg ml$^{-1}$). Forty-eight hours after transfection, cells were seeded at a high dilution to generate single-cell-derived clones. Lysates from HEK293$^{Cas9/KLHL15\Delta}$ cell clones were screened for the loss of KLHL15 expression by western blotting. Finally, the targeted genomic region was amplified by PCR using the following primers (forward, 5'-TGGTGAAGTTCTGCTGC TCT-3'; reverse, 5'-GCCTAGACATGAGTGGCACA-3') and analysed for the presence of insertion and deletion (Indel) mutations by deep-sequencing. All cell lines were confirmed to be free of mycoplasma contamination.

**Cell irradiation.** IR was given using a Faxitron X-ray machine. Laser micro-irradiation was performed as described previously[64]. Briefly, 24 h before irradiation, cells were grown in medium supplemented with 10 µM BrdU. Cells were microirradiated using a MMI CELLCUT system containing a UVA laser of 355 nm (Molecular Machines and Industries, Zurich, Switzerland). The laser intensity was set to 50% energy output and each cell was exposed to the laser beam for <300 ms.

**Chemicals and peptides.** MG-132, cycloheximide and CPT were purchased from Sigma. MLN-4924 was purchased from Active Biochem. The biotinylated 15-mer peptide encompassing CtIP amino acid residues 830–844 was purchased from Bachem AG (Bubendorf, Switzerland) and dissolved in water at 1 mg ml$^{-1}$.

**Vectors.** GFP or FLAG-tagged CtIP expression constructs and pGEX-4T1 plasmids for bacterial expression of CtIP fragments were described previously[8].

The pCMV6-myc-FLAG-KLHL15 expression plasmid was obtained from Origene (NM_030624 Human cDNA True ORF Clone, Origene). The pcDNA3.1-6xHis-Ubiquitin as well as the pcDNA3.1-HA-KLHL22 expression plasmid[65] were a gift from Matthias Peter (ETH Zurich, Switzerland). All CtIP and KLHL15 mutations were introduced by site-directed mutagenesis using Expand Long Template PCR System (Roche) and confirmed by sequencing. Plasmids were transfected either by using the standard calcium phosphate method or FuGENE 6 (Promega) according to manufacturer's instructions.

**Antibodies.** A complete list of all primary antibodies (with manufacturers, catalogue numbers and applications) used throughout this study can be found in Supplementary Table 1. Secondary HRP-conjugated anti-mouse and anti-rabbit antibodies were from GE-Healthcare and the HRP-conjugated anti-goat antibody was from Santa Cruz Biotech. Alexa Fluor-488, -594, and -647-conjugated secondary antibodies were purchased from Invitrogen. Rabbit polyclonal antibodies specific for KLHL15 were generated as follows. Human KLHL15 cDNA corresponding to amino acids 300–604 was cloned into pET30a (Novagen) vector for expression in *Escherichia coli*. The recombinant His-tagged KLHL15 fragment was purified using Ni-NTA (Qiagen) following the manufacturer's instructions and subsequently used to immunize rabbits. After five immunizations, serum was obtained and purified against the recombinant antigen. For that, 100–200 µg of the KLHL15 antigen was loaded onto SDS–polyacrylamide gel electrophoresis (SDS–PAGE) and then transferred to a nitrocellulose membrane before staining with Ponceau S. The part of the membrane containing the antigen was cut out, blocked with 2% BSA in TBS-T for 1 h and then incubated with the serum overnight at 4 °C. Bound antibodies were eluted with 0.15 M glycine-HCl, pH 2.3. 1 M Tris-HCl, pH 8.8, was immediately added to neutralize the pH of the antibody solution to pH 7.5.

**siRNA.** Transfection of siRNA oligos was done using Lipofectamine RNAiMAX (Invitrogen). CNTL, CtIP, CUL3 and KLHL15#2 were purchased from Microsynth and the sequences (5' to 3') were as follows: CNTL (luciferase; 5'-CGUACGCG GAAUACUUCGA-3')[8], CtIP (5'-GCUAAAACAGGAACGAAUC-3')[8], CUL3 (5'-CAACACTTGGCAAGGAGAC-3')[66] and KLHL15#2 (5'-GCGTAAACATCG AGGGAG-3'). SMARTpool ON-TARGETplus Human KLHL15 siRNA (KLHL15#1) was purchased from Dharmacon. Trisilencer-27 human KLHL15 siRNA B targeting the 3'-untranslated region of KLHL15 (KLHL15#3) was purchased from OriGene.

**Double affinity purification coupled to mass spectrometry.** The procedure was performed as described previously with some minor modifications[67]. Briefly, CtIP cDNA was subcloned into the pN-TGSH plasmid (Dualsystems Biotech AG, Schlieren, Switzerland) for tetracycline (Tet)-inducible expression of strep-hemagglutinin (SH)-tagged CtIP bait protein. An isogenic cell line was generated using Flp-recombinase-mediated recombination through single FRT sites present in the pN-TGSH-CtIP expression construct and the genome of Flp-In HEK293 cells (Invitrogen) stably expressing the Tet repressor. After transfection, HEK293$^{SH-CtIP}$ cells were selected on hygromycin for 2 weeks, tested for Tet-inducible expression of SH-CtIP and used for subsequent double affinity purification. The affinity-purified proteins were digested into peptides and the peptide mixture was separated on a C18 HPLC column. Mass spectrometry analysis (direct liquid chromatography-tandem mass spectrometry) was performed using an LTQ Orbitrap XL mass spectrometer (Thermo Fisher Scientific). Peptides of only three proteins, CtIP (Bait), KLHL15 and Cullin-3, were identified in both biological replicates (see Fig. 1a).

**RNA extraction and real-time quantitative RT–PCR.** Total RNA was extracted using the GenElute Mammalian Total RNA Miniprep Kit (Sigma) according to the manufacturer's protocol. Reverse transcription of mRNA was carried out using the Transcriptor First Strand Synthesis kit (Roche). KLHL15 mRNA expression analysis was performed using LightCycler 480 SYBR Green I Master (Roche) and the following primers: KLHL15 (Fw: 5'-ATCATTCAGAATATCCGGTTTT GCT-3', Rw: 5'-TTCAATGCTTGGTCAACTTCGTAA-3') and PBGD (Fw: 5'-CA ACGGCGGAAGAAAACAG-3', Rw: 5'-TCTCTCCAATCTTAGAGAGTG-3'). PBGD expression served as an internal control for quantitative RT-PCR assays and was used to normalize KLHL15 expression levels.

**Purification of recombinant human KLHL15.** The KLHL15 gene was amplified from pCMV6-KLHL15 vector using PCR and subcloned into the pFB-MBP-fusion plasmid (a kind gift of Petr Cejka). The virus was produced using a Bac-to-Bac system (Invitrogen) according to manufacturers' recommendations. MBP-KLHL15 was purified as described previously[68]. Briefly, pellets of 3.2 liters of cultured Sf9 cells expressing MBP-KLHL15 were resuspended in three pellet volumes of lysis buffer (50 mM Tris-HCl, pH 7.5, 1 mM DTT, 1 mM EDTA, cOmplete EDTA-free Protease Inhibitor Cocktail tablets (Roche), 1 mM PMSF and 30 µg ml$^{-1}$ leupeptin). Cells were incubated for 15 min with gentle agitation, and then two pellet volumes of ice-cold 50% glycerol were added to the sample. Next, 5 M NaCl (6.5% of the total solution volume) was added dropwise to the sample, and the

solution was incubated for 30 min with gentle agitation. The soluble extract was obtained by pelleting the insoluble material at 20,000g for 30 min. The soluble cleared extract was bound to pre-equilibrated amylose resin (New England Biolabs). The resin was extensively washed with wash buffer (50 mM Tris-HCl, pH 7.5, 5 mM β-mercaptoethanol, 250 mM NaCl, 10% glycerol, 1 mM PMSF, 10 ug ml$^{-1}$ leupeptin). The resin was incubated for an additional 10 min in wash buffer supplemented with 2 mM ATP and 10 mM MgCl2 to remove some contaminating chaperones. MBP-KLHL15 was eluted with elution buffer (50 mM Tris-HCl, pH 7.5, 5 mM β-mercaptoethanol, 250 mM NaCl, 10% glycerol, 1 mM PMSF, 10 ug ml$^{-1}$ leupeptin and 10 mM maltose). Next, the maltose-binding protein (MBP) was cleaved by PreScission protease and the sample was applied on Ni-NTA resin (Qiagen). The resin was washed with washing buffer containing 40 mM imidazole and eluted in the same buffer containing 400 mM imidazole. Pooled fractions were dialysed against dialysis buffer (50 mM Tris-HCl, pH 7.5, 5 mM β-mercaptoethanol, 300 mM NaCl, 10% glycerol, 0.5 mM PMSF). The sample was aliquoted, frozen in liquid nitrogen and stored at −80 °C.

**CellTiter-Blue Cell Viability assay.** U2OS Flp-In T-REx cells stably expressing doxycycline (Dox)-inducible siRNA-resistant forms of GFP-CtIP or GFP-KLHL15 were transfected with the indicated siRNA oligos. Twenty-four hours post-transfection, cells were seeded in triplicates at a density of 500 cells/well in 96-well plates in medium supplemented with Dox (1 μg ml$^{-1}$). 24 h later, cells were continuously treated with indicated doses of camptothecin and grown for 4 days at 37 °C. To measure cell viability, CellTiter-Blue reagent (Promega) was added on the last day, cells were incubated at 37 °C for 4 h and fluorescence was measured at 560/590 nm.

**Colony-formation assay.** U2OS Flp-In T-REx cells were induced to express GFP-KLHL15-wt or GFP-KLHL15-Y552A using Dox (1 μM). Cells were either mock-treated (DMSO) or treated with the indicated doses of CPT 24 h after Dox-induction. HEK293$^{Cas9}$ wt or *KLHL15Δ* cells were seeded on poly-lysine coated plates and 24 h later irradiated with 1, 2 or 4 Gy. Cells were cultured for 10 days at 37 °C. Colonies were stained with a crystal violet/ethanol (0.5%/20%) solution and counted.

**Immunoblotting and triton extraction.** If not specified otherwise, cell extracts were prepared in Laemmli buffer (4% SDS, 20% glycerol, 120 mM Tris-HCl pH 6.8). Proteins were resolved by SDS–PAGE and transferred to nitrocellulose. Immunoblots were performed with appropriate antibodies and proteins visualized using the Advansta WesternBright ECL reagent and the VilberLourmat Fusion Solo S imaging system. Isolation of Triton-insoluble (chromatin-enriched) fractions was performed as previously described[39]. Briefly, cells were rinsed twice in cold PBS, incubated for 5 min on ice in pre-extraction buffer (25 mM HEPES pH 7.4, 50 mM NaCl, 1 mM EDTA, 3 mM MgCl2, 300 mM sucrose, 0.5% Triton X-100 and protease inhibitors). After buffer removal, adherent cellular material was harvested by scraping the cells into Laemmli buffer. The chromatin-enriched fraction was then heat denatured, sonicated and analysed by immunoblotting. Uncropped immunblots are shown in Supplementary Fig. 9.

**Pull-down assays.** HeLa nuclear extracts were purchased from CilBiotech (Mons, Belgium). For all pull-down assays, cells were lysed in NP-40 extraction buffer (50 mM Tris-HCl, pH 7.5, 120 mM NaCl, 1 mM EDTA, 6 mM EGTA, 15 mM sodium pyrophosphate and 1% NP-40 supplemented with phosphatase inhibitors (20 mM NaF, 1 mM sodium orthovanadate) and protease inhibitors (1 mM benzamidine and 0.1 mM PMSF) and clarified by centrifugation. For GST pull-down assays, GST fusion plasmids were grown in BL21 RIL (CodonPlus) *E. coli* (Stratagene) and recombinant proteins were expressed by incubating the bacteria for 24 h at 16 °C after the addition of 100 μM IPTG. After centrifugation, the bacterial pellet was resuspended in cold PBS, supplemented with 1% Triton X-100 and protease inhibitors (1 mM PMSF, 1 mM benzamidine, and Roche protease inhibitor cocktail). After sonication and centrifugation, GST-tagged proteins were purified from soluble extracts using Glutathione Sepharose 4 Fast Flow beads (GE Healthcare). GST fusion proteins bound to glutathione beads were mixed with 1 mg of HeLa nuclear extract or 1 mg of HEK293 cells extracts and incubated for 1 h at 4 °C in 1 ml of PBS-1%Triton. Beads were then washed three times with NTEN300 buffer (0.5% NP-40, 0.1 mM EDTA, 20 mM Tris-HCl pH 7.4 and 300 mM NaCl) and once with TEN100 (20 mM Tris-HCl pH 7.4, 0.1 mM EDTA and 100 mM NaCl) buffer. Recovered complexes were boiled in SDS sample buffer and analysed by SDS–PAGE followed by immunoblotting. For MBP pull-down assays, Sf9 soluble cleared extract expressing MBP-KLHL15-His were incubated with amylose resin (New England Biolabs) and washed four times with NTEN300 buffer (0.5% NP-40, 0.1 mM EDTA, 20 mM Tris-HCl pH 7.4, 300 mM NaCl) and once with TEN100 buffer (20 mM Tris-HCl pH 7.4, 0.1 mM EDTA and 100 mM NaCl). MBP fusion proteins bound to amylose beads were mixed with 1 mg of HEK293T lysates overexpressing GFP-CtIP and incubated for 1 h at 4 °C. Beads were then washed four times with NTEN300 buffer and once with TEN100 buffer, complexes were boiled in SDS sample buffer and analysed by SDS–PAGE followed by immunoblotting. For peptide pull-down assays, biotinylated CtIP peptides were incubated with Dynabeads streptavidin resin (Qiagen) and washed

four times with 1%Triton-PBS. Beads were then mixed with 2.5 mg of U2OS or 1 mg of HEK293T lysates and incubated for 1 h at 4 °C. Beads were then washed four times with NP-40 extraction buffer, complexes were boiled in SDS sample buffer and analysed by SDS–PAGE followed by immunoblotting.

**Co-immunoprecipitation.** For FLAG and HA co-IP assays, cells were lysed in NP-40 extraction buffer, sonicated and clarified by centrifugation. A total of 1–3 mg of lysates were incubated with anti-FLAG M2 or anti-HA affinity resin (Sigma) for 2 h or overnight at 4 °C. Beads were then washed four times with 1xTNE-T (50 mM Tris-HCl, pH 7.5, 140 mM NaCl, 5 mM EDTA and 1%Triton) and once with 1 × TNE (50 mM Tris-HCl, pH 7.5, 140 mM NaCl and 5 mM EDTA), complexes were boiled in SDS sample buffer and analysed by SDS–PAGE followed by immunoblotting. For GFP co-IP assays, cells were lysed in GFP-IP buffer (100 mM NaCl, 0.2% NP-40, 1 mM MgCl$_2$, 10% glycerol, 5 mM NaF, 50 mM Tris-HCl, pH 7.5) supplemented with phosphatase inhibitors (20 mM NaF, 1 mM sodium orthovanadate) and with protease inhibitors (1 mM benzamidine and 0.1 mM PMSF) and incubated with Benzonase (Novagen) for at least 30 min at 4 °C. After Benzonase digestion, the NaCl and ETA concentrations were adjusted to 200 and 2 mM, respectively, and lysates cleared by centrifugation. Four milligram of lysates were incubated with GFP-Trap agarose beads (ChromoTek) previously blocked with 5% BSA in GFP-IP buffer (with 200 mM NaCl and 2 mM EDTA final concentrations) for 1 h at 4 °C. Beads were then washed five times with GFP-IP buffer (with 200 mM NaCl and 2 mM EDTA final concentrations), complexes were boiled in SDS sample buffer and analysed by SDS–PAGE followed by immunoblotting.

***In vivo* ubiquitination assay.** Stable HEK293 Flp-In T-REx GFP-CtIP cells were transfected with siRNA oligos (20–40 nM). Forty-eight hours later, cells were transfected with His-ubiquitin and GFP-expression was simultaneously induced with 1 μg ml$^{-1}$ Dox. Eight hours after induction a second transfection with siRNA oligos was performed. Alternatively, HEK293T cells were cotransfected with His-Ubiquitin and GFP-CtIP expression constructs. Twenty-four hours after induction or transfection of GFP-CtIP, respectively, cells were treated for 4–6 h with 20 μM MG-132 and then washed and scraped in 500 μl of ice-cold PBS. 2% of the cell suspension was used for direct immunoblot analysis. The remaining cells were lysed in Buffer A (6 M guanidine–HCl, 0.1 M Na$_2$HPO$_4$/NaH$_2$PO$_4$ pH 8.0, 10 mM imidazole), and lysates were incubated with Ni-NTA agarose beads for 3 h under rotation at room temperature. The beads were washed two times with Buffer A, two times with Buffer A/TI (one volume Buffer A: three volume Buffer TI (25 mM Tris-HCl pH 6.8, and 20 mM imidazole)), and three times with Buffer TI. Proteins were eluted from the beads by boiling in 2 × SDS sample buffer supplemented with 250 mM imidazole and analysed by immunoblotting.

***In vitro* ubiquitination assay.** Recombinant E2 enzymes (UbcH5b and Cdc34) were purified as described before. In brief, UbcH5b and Cdc34 full-length constructs were expressed as GST-fusions in BL21 *E. coli* and purified over a 5 ml GST HiTrap column (GE Healthcare) and a Superdex75 size exclusion chromatography column (GE Healthcare) using standard protocols[69,70]. Neddylated CUL3 in complex with RBX1 was produced as described before. In brief, full-length CUL3, tagged with an N-terminal PreScission-cleavable StrepII2x tag, and RBX1 were cloned into a pFBDM multibac transfer vector. Following the generation of recombinant baculoviruses, CUL3/RBX1 complexes were expressed and purified from high five insect cells following standard Strep-Tactin affinity, ion exchange and Superdex200 size exclusion chromatography procedures. Neddylation of CUL3/RBX1 was performed by incubating 10 μM CUL3/RBX1, 0.5 μM APPB1/UBA3 (NEDD8-specific activating enzyme E1), 1 μM UBC12 (NEDD8-specific conjugating enzyme E2) and 20 μM NEDD8 at room temperature for 10 min in buffer containing 100 mM Tris-HCl pH 7.6, 100 mM NaCl, 2.5 mM MgCl$_2$ and 150 mM ATP. The neddylated CUL3-N8/RBX1 was subsequently purified by Strep-Tactin affinity chromatography with the untagged neddylation reagents retained in the flow through[30]. Full-length wild-type human KLHL21 was cloned into a pGEX-5 expression vector and expressed in BL21 *E. coli*. The cells were lysed by sonication in the presence of protease inhibitors and lysozyme. The soluble supernatant was bound to a 5 ml GST HiTrap column (GE Healthcare) with a flow rate of 1 ml min$^{-1}$. The column was washed with 20 column volumes (CV) of washing buffer and eluted with five CV of washing buffer supplemented with 10 mM reduced L-glutathione (AppliChem). The eluted GST-KLHL21 was incubated in a 1:50 molar ratio with GST-tagged PreScisscion protease at 4 °C and the buffer exchanged by dialysis into 150 mM NaCl, 20 mM HEPES pH 7.6, 10% glycerol, 2 mM DTT. To remove cleaved GST and GST-PreScission proteins, the sample was run through a GST HiTrap column a second time, and purified KLHL21 was collected in the flow-through. The CtIP substrate was purified as follows: NP-40 lysates of HEK293T cells transfected with FLAG-CtIP were incubated with anti-FLAG M2 beads (Sigma) for 1 h at 4 °C. The resin was washed four times with NTEN500 buffer (0.5% NP-40, 0.1 mM EDTA, 20 mM Tris-HCl, pH 7.4, 500 mM NaCl) and once with TEN100 buffer. FLAG-CtIP was eluted with 3xFLAG peptides (Sigma) in TBS. If not indicated otherwise, ubiquitination reactions were performed at 37 °C for 90 min with 400 nM of FLAG-CtIP, 250 nM of KLHL15 or KLHL21, 200 nM of neddylated

CUL3/RBX1, 250 nM UbE1 (Lifesensor), 500 nM UbcH5b or 2 μM Cdc34 and 75 μM ubiquitin wild-type or mutants (Bostonbiochem). The reaction was stopped by the addition of SDS sample buffer supplemented with 20 mM DTT and boiling for 5 min at 95 °C and analysed by SDS–PAGE followed by immunoblotting on PVDF membrane (Amersham).

**Immunofluorescence microscopy.** U2OS cells grown on coverslips were fixed in 4% formaldehyde (w/v) in PBS for 15 min and permeabilized with Triton X-100 (0.1% in PBS) for 5 min at room temperature. After incubation with indicated primary and appropriate Alexa Fluor-488, -594 and -647 conjugated secondary antibodies (1:1,000) (Life Technologies) for 30 min, coverslips were mounted with Vectrashield (Vector Laboratories) containing DAPI and sealed. Images were acquired on a Leica DMRB fluorescence microscope.

**Flow cytometry-based resection assay.** U2OS Flp-In T-REx cells stably expressing Dox-inducible siRNA-resistant forms of GFP-CtIP were transfected with CtIP siRNA. 24 h post-transfection, Dox (1 μg ml$^{-1}$) and BrdU (30 μM) were added to the cells. Forty-eight hours post-transfection, cells were either mock-treated or treated with CPT (1 μM). Alternatively, U2OS Flp-In T-REx cells stably expressing Dox-inducible GFP-KLHL15 were induced with Dox (1 μg ml$^{-1}$) and after 24 h treated with CPT (1 μM). Alternatively, HEK293$^{Cas9}$ cells were directly treated with CPT (1 μM). 1 h following CPT treatment, all cells were collected, pre-extracted with Triton-X100 (0.3% in PBS) for 15 min on ice and fixed with 4% formaldehyde for 10 min at room temperature. Cells were then incubated with antibodies against RPA2 or BrdU for 1 h at room temperature, stained with Alexa-647 conjugated secondary antibodies (1:250) for 30 min at room temperature and counterstained with DAPI/RNase. Samples were analysed by flow cytometry on a CyAn ADP 9 (Dako). Data analysis was performed using FlowJo X software (Tree Star).

**GFP reporter assays.** DSB repair efficiency by HR or NHEJ was measured in DR-GFP or EJ5-GFP HEK293 cell lines as described previously[28]. Briefly, $0.6 \times 10^6$ cells were plated in poly-L-lysine-coated six-well plates and transfected with siRNA oligos (20–40 nM). The next day, $0.22 \times 10^6$ cells were reseeded in 12-well plates. Forty-eight hours after siRNA transfection, cells were either mock-transfected or transfected with 0.6 μg of the I-SceI expression plasmid (pCBASce) mixed with 0.2 μg of the indicated siRNA-resistant FLAG-CtIP (pcDNA3) constructs. Four hours after plasmid transfection, the medium was replaced and a second transfection with siRNA oligos was performed. Alternatively, GFP reporter assays upon siRNA-mediated knockdown conditions only were performed like described above except that the cells were either mock-transfected or transfected with 0.6 μg of the I-SceI expression plasmid. Alternatively, $0.22 \times 10^6$ cells were seeded in 12-well plates and 24 h later either mock-transfected or transfected with 0.6 μg I-SceI expression plasmid mixed with 0.2 μg of the indicated FLAG-KLHL15 constructs (pCMV6). Forty-eight hours after I-SceI transfection, cells were analysed for GFP expression by flow cytometry on a CyAn ADP 9 (Dako).

**Data availability.** The authors declare that all remaining data supporting the findings of this study are contained within the article and its Supplementary Information Files or available from the author upon request.

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

## Acknowledgements

We are very grateful to Josef Jiricny, Pavel Janscak, Stefano Ferrari, Petr Cejka and Massimo Lopes (Institute of Molecular Cancer Research, University of Zurich, Switzerland) for sharing reagents and to Daniel Durocher (The Lunenfeld-Tanenbaum Research Institute, Mount Sinai Hospital, Toronto, Canada) for providing U2OS Flp-In T-REx cells. We thank Izabela Sumara (Institute of Genetics and Molecular and Cellular Biology, University of Strasbourg, France) for critical reading of the manuscript. R.I.E. was supported by a Marie-Curie Fellowship, and the Peter laboratory by grants of the Swiss National Science Foundation, the European Research Council and the ETH Zurich. R.F. is supported by a grant from the Spanish Ministry of Economy and Competitiveness (SAF2013-49149-R). This work was mainly supported by grants from the Swiss National Science Foundation (31003A_135507 and 31003A_156023), the Promedica Stiftung and the Vontobel Foundation (to A.A.S.).

## Author contributions

A.A.S. and L.P.F. conceived the original ideas for this study, analysed the data and wrote the manuscript together with A.P. L.P.F. performed all experiments with the help of S.-F.H., A.T., C.W., C.v.a., A.E., O.M. and R.I.E. R.I.E. and M.P. contributed reagents and helped in the experimental design and interpretation of the *in vitro* ubiquitination assays. R.F. generated the anti-KLHL15 antibody.

## Additional information

**Competing financial interests:** The authors declare no competing financial interests.

