## [Peer review file · Nature Communications]

Reviewers' Comments:

Reviewer #1 (Remarks to the Author)

In this manuscript the authors show that the cullin-3 E3 ligase substrate adaptor KLHL15 interacts with CtIP and targets it for ubiquitination and degradation.

The major problems are summarized below:

1. Figure 1c shows that KLHL15 binds CtIP depending on both its BTB-BACK and Kelch domains. Does CtIP specifically bind KLHL15? The authors need to use other Cul3 substrate adaptors as control. To confirm that CtIP and Cul3-KLHL15 forms a trimeric complex, the authors should perform sequential IPs. Finally, the binding needs to be confirmed with endogenous proteins.
2. In figure 1d, the N132A mutant lost the binding with Cul3, so, it should not mediate the degradation of CtIP. Why does the N132A mutant induce a decrease in CtIP protein levels? The half-life of CtIP should be measured after expression of WT and the N132A mutant.
3. In figure 2a, the Y552A mutant is expressed less than the WT, so, it is difficult to make a conclusion on its activity. Again, the half-life of CtIP should be measured after expression of WT and the Y552A mutant.
4. Since an antibody for WB is available, in figure 2d, 2e and 2f, the authors should show KLHL15 protein levels to confirm that the siRNA to KLHL15 worked.
5. Figure 2d: The siRNA #3 is much less efficient than #1. However, #3 siRNA induces the same effect on the CtIP protein level, as #1. How is this possible?
6. Figure 2f: KLHL15 siRNA dramatically increased the protein level of CtIP, however, the half-life is only slightly increased. How reproducible is this result? The authors should quantify the half-life in several experiments.
7. Figure 2g and 2h showed that KLHL15 promoted the ubiquitination of CtIP in vivo and in vitro, is it formed K48 ubiquitin chain? A negative control with another ubiquitin ligase would be advisable.
8. Figure 3c: The authors need to show a long and short exposure of the CtIP blot that allows a direct comparison of the effect on the half-life.
9. Figure 4b: Again, the Y552A mutant expression level is much less than the WT, so, the phenotype might be caused by the different expression level of KLHL15, not the Cul3 binding.
10. In figure 5b, the authors showed that delta C1 mutant lost the binding with KLHL15. This could be due to a change in localization (the C-terminal domain is required for DNA-end resection) and needs to be verified.
11. Fig. 6d: the mutants seem less ubiquitinated, but also less precipitated (i.e. less protein is pulled down), which could be the real reason for the apparent decrease in ubiquitination. The experiment needs to be normalized.
12. Figure 6e: Although the Y842A mutant expresses more than the WT when co-expressed with KLHL15, it doesn't mean that the Y842A mutant is more stable. A measurement of the half-life is essential.

Reviewer #2 (Remarks to the Author)

CtIP is a key component of the homologous recombination mechanism through its effect on DNA end resection. In this manuscript the authors clearly shows that KLHL15 controls CtIP protein levels by promoting ubiquitin-dependent degradation. The results are very clear, the experimental design very neat and very well executed and I have no major concerns in respect the authors main conclusions. Having said that, I am not sure that this represents a significant enough step forward on our knowledge to grant publication in Nature Communications. We know that CtIP is essential for DNA end resection, thus it is not surprising that the more CtIP protein there is the more resection you have and that reducing CtIP levels lead to impaired resection. Hence, the information here is only incremental, and might be relevant only for a minority of scientist interested in CtIP regulation. There is no evidence whatsoever in the manuscript that this is part of a regulatory mechanism that reacts to something. Indeed, it seems that is just a constitutive

response. In that regard, does the interaction between KLHL15 and CtIP change upon DNA damage or depending on cell cycle?

It is clear that controlling CtIP levels is important, and probably that is why several ubiquitin ligases have been described to target CtIP (some found by the same lab is submitting this paper). The idea that this might be connected with their previous observation of PIN1 role is interesting, but it should be address in more detail to make this manuscript suitable for this journal. The authors have all the tools to test this idea, maybe by combining the CtIP-S276A/T315A mutant with KHLH15 overexpresion or downregulation.

Regardless of the fate of this manuscript, there are some points I would like to raise to strengthen the story:

1. The connection with Ubiquitination/neddylolation by using the inhibitors is a good starting point, but not strong enough to state that KLHL15 favors Neddylated-Cul3-mediated ubiquitin ligation. The problem is that MG132 or MLN494 stabilizes CtIP in both KLHL15 overexpressing and not overexpressing cells (Figure 2b and 2c). In vitro it is clear (figure 2h). In vivo it will help if they repeat the overexpression of KLHL15 in cells depleted for Cul3.
2. Does overexpression of CtIP behave like KHLH15 depleted or KO cells? Easy to test by comparing GFP and GFP-CtIP transfected cells in the presence of endogenous CtIP.
3. Assays after DNA damage have been performed with a drug, CPT that causes DSBs in S phase. Thus, it repaired exclusively by recombination. What happens when the authors test other types of DNA damage that can be repaired by NHEJ also? Are cells overexpressing or depletion KHLH15 also affected the same way? Are they more sensitive to other types of DNA damage?
4. Related, I miss in several figures (4 and 7 mainly) a positive control, preferentially CtIP depletion. It will help to establish the degree in which the repair is compromised.
5. CHX assays should be quantified. As the starting point is different, and Western blot signal saturate, it will help to strengthen the message.
6. Figure 6c, a proper loading control (not Ponceau staining) is desirable.
7. Figure 8 does not add any information. The model is simple enough to be explained without a cartoon. I suggest to eliminate this figure.

Response to referees:

We would like to thank both reviewers for their overall constructive criticism on our study. Please find our detailed point-to-point response to each of the reviewer's comments below.

In the revised manuscript, we have now included the following new data:

- 1. To further corroborate our main finding that CtIP forms a complex with CUL3-KLHL15, we show that wild-type CtIP, but not CtIP-Y842A single point mutant, co-precipitates endogenous KLHL15 and CUL3. Moreover, a biotinylated 15-mer peptide encompassing residues 830-844 and containing the conserved 'FRY motif' specifically pulls-down CUL3 and KLHL15 from cells.*
- 2. Using two additional KLHL15 mutants in the 3-box (I136A) and the Kelch domain (G386C) in immunoprecipitation experiments, we further demonstrate that KLHL15 is a CUL3 substrate adaptor, linking CtIP to the ubiquitin-proteasome pathway.*
- 3. Quantification of cycloheximide chase experiments performed in various conditions (siRNA, KLHL15 knockout cells, KLHL15 mutant cells) establishes that KLHL15 controls CtIP protein turnover. In addition, we show that CtIP-Y842A mutation results in a prolonged protein half-life and protein accumulation.*
- 4. Using in vitro ubiquitination assays with ubiquitin mutants, we show that CUL3-KLHL15 E3 ligase catalyzes the formation of K48-linked ubiquitin chains on CtIP, supporting our main finding that KLHL15 promotes proteasomal degradation of CtIP.*
- 5. Consistent with our main finding that KLHL15 regulates DSB repair pathway choice, we now include clonogenic survival assays showing that KLHL15 knockout cells are hypersensitive to ionizing radiation, most likely caused by excessive DNA-end resection and suppression of NHEJ.*
- 6. Based on a series of new experiments, we conclude that PIN1-mediated CtIP isomerization (Steger et al, Molecular Cell, 2013) facilitates CtIP ubiquitination by CUL3-KLHL15.*

Reviewer #1:

In this manuscript the authors show that the cullin-3 E3 ligase substrate adaptor KLHL15 interacts with CtIP and targets it for ubiquitination and degradation. The major problems are summarized below:

- 1. Figure 1c shows that KLHL15 binds CtIP depending on both its BTB-BACK and Kelch domains. Does CtIP specifically bind KLHL15? The authors need to use other Cul3 substrate adaptors as control. We think that our data is consistent with the idea that KLHL15 is indeed the substrate adaptor which specifically recruits CtIP to the CUL3 E3 Ligase complex. To further corroborate this conclusion, we have now included new data showing that KLHL22, another CUL3 substrate adaptor does not interact with CtIP (Supplementary Figure 1c).*

To confirm that CtIP and Cul3-KLHL15 forms a trimeric complex, the authors should perform sequential IPs. Finally, the binding needs to be confirmed with endogenous proteins.

This is of course a very valid point and we made several efforts to address it. Nevertheless, we were unable to detect endogenous KLHL15-CUL3-CtIP complexes by conventional co-immunoprecipitation experiments. This could be explained by the fact that our anti-KLHL15 antibody was raised against an epitope mapping between amino acids 300-604. As this region encompasses the Kelch domain responsible for substrate binding, we concluded that the co-IP of endogenous CtIP mainly failed because the KLHL15 antibody masks the interaction site for CtIP. Similarly, the anti-CtIP antibodies at our disposal were raised against the C-terminal region of CtIP, again competing for KLHL15 binding. We also failed to co-precipitate endogenous CtIP using anti-CUL3 antibody, most likely because CUL3 interacts with various different substrate adaptors. Therefore, only a small proportion of CUL3 exists temporally in a complex with KLHL15-CtIP, which may be below detection levels. Finally, enzyme-substrate interactions are commonly known to be very transient and intrinsically difficult to detect by co-IPs.

However, we have included new results showing that endogenous KLHL15 and CUL3 are efficiently co-immunoprecipitated with GFP-CtIP-wt, but not with GFP-CtIP-Y842A (Figures 1b and 6c). Moreover, we already showed that endogenous CtIP and CUL3 readily interact with FLAG-KLHL15 expressed in cells and now further substantiate these findings using two additional KLHL15 mutants (Figures 1d-f). Finally, we present new evidence that a short 15-mer CtIP peptide harboring the FRY KLHL15 docking motif efficiently pulls down endogenous CUL3-KLHL15 complex from parental HEK293 cells but not from HEK293 KLHL15 knockout cells (Fig. 5f).

2. In figure 1d, the N132A mutant lost the binding with Cul3, so, it should not mediate the degradation of CtIP. Why does the N132A mutant induce a decrease in CtIP protein levels? The half-life of CtIP should be measured after expression of WT and the N132A mutant.

We agree with the reviewer that data shown in Figure 1d is not fully in line with the concept that KLHL15-CUL3 targets CtIP for degradation. Therefore, we have repeated the co-IP experiment with FLAG-tagged KLHL15 '3-box' mutants (N132A and I136A) several times and realized that the interaction with CUL3 is not as strongly compromised as previously indicated. We apologize for this mistake and have now replaced the previous Figure with a more representative immunoblot (Figure 1e). This new data may also at least partially explain why KLHL15-N132A mutant is only partially defective in promoting CtIP degradation. As requested by the reviewer, we performed several cycloheximide chase experiments and find that CtIP turnover rate is indeed comparable between KLHL15-N132A and KLHL15-wt, whereas KLHL15-Y552A significantly increased CtIP protein half-life (Rebuttal Figure 1 and Supplementary Figure 1f).

Rebuttal Figure 1. HEK293T cells transfected with either empty vector (EV) or the indicated FLAG-KLHL15 expression constructs for 48 h were treated with cycloheximide (CHX, 100 µg/ml) for the indicated time points and analysed by immunoblotting.

We have extensively discussed this contradicting issue with our collaborators (Matthias Peter and Radoslav Enchev) and we think that it could be at least in part explained by the molecular architecture of CUL3-based E3 ligase complexes. It is known that most, if not all, CUL3 ligases dimerize through the BTB domain of their adaptor protein and it is proposed that those dimers are engaged in a single complex (Merlet et. al, 2009 and Canning et. al, 2013). The best investigated example of such dimerization is CUL3-Keap1-Nrf2, in which two Keap1 substrate adaptor proteins simultaneously bind to a single Nrf2 target molecule, thus building a complex consisting of two CUL3, two Keap1 and one Nrf2 (Canning et al, 2015). We believe that the same bivalent model can be applied to CUL3-KLHL15-CtIP, in which two KLHL15 subunits are required for the interaction with CtIP. If this is the case, then KLHL15 3-box mutants (N132A and I136A) retain the capacity to heterodimerize with endogenous wild-type KLHL15 (present in our cells), which in turns interacts with CUL3 and ensures the formation of a functional catalytic center still able to ubiquitinate CtIP. In contrast, as both KLHL15 subunits are required for CtIP binding, Kelch repeat domain mutants (G386C or Y552A) more potently abolish the interaction with CtIP and ultimately, the degradation of CtIP (as indicated in Figure 1f and Supplementary Figure 1f). However, further detailed structural investigations are needed to experimentally support this model, which in our view are beyond the scope of this study.

3. In figure 2a, the Y552A mutant is expressed less than the WT, so, it is difficult to make a conclusion on its activity. Again, the half-life of CtIP should be measured after expression of WT and the Y552A mutant.

We have analyzed the half-life of endogenous CtIP in U2OS cells and compared it to cells expressing either KLHL15-wt or Y552A (Rebuttal Figure 2). However, CHX treatment of U2OS cells does not lead to a reduction in CtIP protein levels. We reasoned that this is most likely due to very low KLHL15 expression levels in U2OS, thereby rendering CtIP much more stable as compared to the situation in HEK293(T) cells (compared to point 4). Nevertheless, when we induced the expression of KLHL15 in the stable U2OS cells, CtIP protein half-life is increased in Y552A compared to wild-type cells (Rebuttal Figure 2).

Rebuttal Figure 2. U2OS Flp-In T-REx cells inducibly expressing GFP-KLHL15-wt or GFP-KLHL15-Y552A were cultivated in presence or absence of Dox (1 µg/ml). 24 h post-induction cells were treated with cycloheximide (CHX, 100 µg/ml) for the indicated time points and analysed by immunoblotting.

For these reasons and for clarity, we have decided to exclude this experiment from the revised manuscript. However, as already outlined above, we performed CHX chase experiments and measured CtIP half-life in HEK293T cells transiently transfected with KLHL15-wt and KLHL15-Y552A (Supplementary Figure 1f). Lastly, we decided to exclude Figure 2a from the revised manuscript, as we have now more extensively characterized U2OS GFP-KLHL15-Y552A stable clones in the revised manuscript (Figure 4 and Supplementary Figures 4b-c). Concerning KLHL15-Y552A expression levels in stable U2OS cells, please also refer to point 9.

- Since an antibody for WB is available, in figure 2d, 2e and 2f, the authors should show KLHL15 protein levels to confirm that the siRNA to KLHL15 worked.

We apologize for these omissions. As shown in Figure 2d, we could confirm the knockdown of KLHL15 by western blot analysis. However, since the amount of endogenous KLHL15 is extremely low in U2OS cells, we could only detect KLHL15 in the chromatin-enriched fraction. For the same reason, we are unable to show KLHL15 immunoblot in Figure 2e. However, we believe that the GFP immunoblot is sufficient to verify efficient silencing of KLHL15 expression by siRNA oligos. Finally, concerning KLHL15 protein levels in Figure 2f (moved to Supplementary Figure S2c in the revised manuscript), we now include a KLHL15 immunoblot confirming partial downregulation of KLHL15 in HEK293T cells.

- Figure 2d: The siRNA #3 is much less efficient than #1. However, #3 siRNA induces the same effect on the CtIP protein level, as #1. How is this possible? The simplest way for us to answer this valid question is that KLHL15 protein levels are not proportional to KLHL15 mRNA transcript levels, as measured in this experiment (moved to Supplementary Figure 2A). In other words, KLHL15 mRNA turnover may be much shorter than KLHL15 protein turnover and, although mRNA levels upon siRNA#1 transfection are approximately 2-fold lower than upon siRNA#3 transfection, KLHL15 protein levels are similarly reduced with both siRNA oligos. In fact, our new western blot data reveals that the all KLHL15 siRNA oligos used are downregulating KLHL15 (Figure 2d). Still, we agree that there is more residual KLHL15 protein left with siRNA#3. Perhaps, because of the dimeric structure of CRL3s (see point 1

above), this amount of KLHL15 is still not sufficient to promote efficient CtIP ubiquitination and proteasomal degradation.

6. Figure 2f: KLHL15 siRNA dramatically increased the protein level of CtIP, however, the half-life is only slightly increased. How reproducible is this result? The authors should quantify the half-life in several experiments.
This is an important point. We have now independently reproduced this CHX chase experiment several times and included the quantification data in a new Supplementary Figure 2c.
7. Figure 2g and 2h showed that KLHL15 promoted the ubiquitination of CtIP in vivo and in vitro, is it formed K48 ubiquitin chain? A negative control with another ubiquitin ligase would be advisable.
We thank the reviewer for this comment. By using ubiquitin mutants (K48 only and K48R) we now report that CUL3-KLHL15 forms K48-linked ubiquitin chains on CtIP (Supplementary Figure 2e). Moreover, in the same experiment, we included CUL3-KLHL22 E3 ubiquitin ligase as a negative control (Supplementary Figure 2e).
8. Figure 3c: The authors need to show a long and short exposure of the CtIP blot that allows a direct comparison of the effect on the half-life.
We fully agree with the reviewer and modified Figure 3c accordingly. In addition, we have repeated this experiment multiple times and included the quantification data in the same Figure.
9. Figure 4b: Again, the Y552A mutant expression level is much less than the WT, so, the phenotype might be caused by the different expression level of KLHL15, not the Cul3 binding.
We fully agree that this is an important issue. Therefore, we have now included a western blot analysis of three separate stable U2OS clones inducibly expressing GFP-KLHL15-Y552A, clearly demonstrating that regardless of the respective Y552A expression levels, CtIP protein stability and, thus, DNA-end resection activity remains largely unaffected.
10. In figure 5b, the authors showed that delta C1 mutant lost the binding with KLHL15. This could be due to a change in localization (the C-terminal domain is required for DNA-end resection) and needs to be verified.
This is indeed possible. Yet, it was previously shown that GFP-CtIP Δ C1 properly localizes in the nucleus (Sartori et al, 2007). Moreover, we show that full-length GFP-CtIP harboring a single amino acid substitution in the C-terminal domain (Y842A) is defective in KLHL15 interaction, but is still properly localized in the nucleus and efficiently recruited to sites of DNA damage (Supplementary Figures 7b-c).
11. Fig. 6d: the mutants seem less ubiquitinated, but also less precipitated (i.e. less protein is pulled down), which could be the real reason for the apparent decrease in ubiquitination. The experiment needs to be normalized.
In this special type of ubiquitination assay, cells are transfected with His-tagged ubiquitin. Subsequently, ubiquitinated proteins are enriched using a Ni-NTA resin. Ni-NTA eluates are then analyzed by western blot with an

antibody against the protein of interest, in this case GFP-CtIP. Therefore, GFP-CtIP-Y842A is indeed less precipitated because it is less ubiquitinated, even though its expression level is comparable to GFP-CtIP-wt as shown in the input blot (Figure 6f).

12. Figure 6e: Although the Y842A mutant expresses more than the WT when co-expressed with KLHL15, it doesn't mean that the Y842A mutant is more stable. A measurement of the half-life is essential.

We completely agree with the reviewer on this point and thus measured the half-life of GFP-CtIP-wt and Y842A in several independent experiments. The western blot and quantification data clearly indicate that CtIP-Y842A has a prolonged half-life compared to CtIP-wt (Supplementary Figure 7d).

Reviewer #2:

CtIP is a key component of the homologous recombination mechanism through its effect on DNA end resection. In this manuscript the authors clearly shows that KLHL15 controls CtIP protein levels by promoting ubiquitin-dependent degradation. The results are very clear, the experimental design very neat and very well executed and I have no major concerns in respect the authors main conclusions. Having said that, I am not sure that this represents a significant enough step forward on our knowledge to grant publication in Nature Communications. We know that CtIP is essential for DNA end resection, thus it is not surprising that the more CtIP protein there is the more resection you have and that reducing CtIP levels lead to impaired resection. Hence, the information here is only incremental, and might be relevant only for a minority of scientist interested in CtIP regulation. There is no evidence whatsoever in the manuscript that this is part of a regulatory mechanism that reacts to something. Indeed, it seems that is just a constitutive response. In that regard, does the interaction between KLHL15 and CtIP change upon DNA damage or depending on cell cycle?

This is of course a very important point and we have addressed it using several approaches. First, we find that KLHL15 expression is not regulated in a cell cycle-dependent manner (Supplementary Figure 3d). Second, KLHL15 protein levels in HEK293 cells remain unaffected in response to acute CPT treatment (Figure 7d). Third, and this may be interesting, we noticed a slight reduction in the amount of KLHL15, corresponding to a slight increase in the amount of CtIP, in response to high doses of ionizing radiation (Figure 7f). Therefore, it may be possible that KLHL15 protein stability is partially regulated by DNA damage in a spatiotemporal manner, yet depending on the specific type of lesion and the time point following DSB induction. In the case of IR-induced DSBs, one could speculate that KLHL15 initially degrades CtIP to prevent DNA-end resection and facilitate NHEJ, but that at later time points, KLHL15 levels drop so that CtIP is allowed to initiate DNA-end resection and promote HR of the remaining DSBs, which could not have been fixed by NHEJ. Moreover, we now provide new exciting evidence that PIN1-mediated isomerization of CtIP facilitates its ubiquitination by CUL3-KLHL15 (see below).

Furthermore, it has been previously shown that CUL3-Keap1 constitutively targets Nrf2 for degradation under non-stressed conditions and that Nrf2 gets released from the complex in response to oxidative stress (Tian et al, 2012). Therefore, based on our data, we would like to propose a similar mechanism for CtIP, in which CUL3-KLHL15 constitutively targets CtIP for degradation but receives an "off signal" to stop ubiquitinating CtIP in situations where CtIP functions are strictly required to promote cell viability. Moreover, we also envision that KLHL15 expression, and therefore KLHL15-mediated CtIP degradation, may be regulated in a cell- or tissue-specific manner (see discussion for details).

It is clear that controlling CtIP levels is important, and probably that is why several ubiquitin ligases have been described to target CtIP (some found by the same lab is submitting this paper). The idea that this might be connected with their previous observation of PIN1 role is interesting, but it should be address in more detail to make this manuscript suitable for this journal. The authors have all the tools to test this idea, maybe by combining the CtIP-S276A/T315A mutant with KHLH15 overexpression or downregulation.

This is for sure a very important point. Therefore, we have now comprehensively addressed the putative connection between PIN1 and KLHL15 in controlling CtIP turnover in the revised version of the manuscript (see new Figure 8 and Supplementary Figure 8). In brief, we now report that CtIP-S276A/T315A (2A) is indeed partially resistant towards KLHL15 overexpression, whereas the KEN-box mutant of CtIP (K467A), refractory to ubiquitylation by the APC/C^{Cdh1} E3 ligase (Lafranchi et al, 2014), is as susceptible to degradation by KLHL15 as wild-type (Figure 8a). This prompted us to further investigate whether there may be a relationship between KLHL15-dependent ubiquitination and PIN1-mediated CtIP isomerization. Indeed, we find that KLHL15-CtIP complex formation is reduced in PIN1-depleted cells (Figure 8b). However, combined with our data showing that CtIP-2A mutant interacts with MBP-KLHL15 (Supplementary Figure 8a), we conclude that isomerization of CtIP by PIN1 is not strictly required for interaction with KLHL15, but rather indirectly stabilizes the ternary CUL3-KLHL15-CtIP complex. Moreover, we also provide evidence that double knockdown of KLHL15 and PIN1 does not further reduce CtIP ubiquitination compared to the single knockdowns (Supplementary Figure 8b), suggesting that the two enzymes (PIN1 and CUL3-KLHL15) epistatically act on the same substrate (CtIP). Finally, we demonstrate that PIN1-dependent CtIP degradation and defective DNA-end resection after CPT treatment are restored in KLHL15 knockout cells (Figure 8c). Taken together, we propose a model in which isomerization of CtIP by PIN1 reinforces KLHL15-mediated CtIP ubiquitination, leading to a higher degradation of CtIP in certain specific biological settings when CtIP activity needs to be suppressed, such as after successful DNA-end resection initiation or during late S/G2 phases of cell cycle when NHEJ is the predominant pathway for DSB repair (Karanam et al, 2012). Please refer to the main text and discussion for more details.

We hope that this reviewer appreciates our efforts in improving both the novelty and the impact of the revised manuscript.

Regardless of the fate of this manuscript, there are some points I would like to raise to strengthen the story:

1. The connection with Ubiquitination/neddylation by using the inhibitors is a good starting point, but not strong enough to state that KLHL15 favors Neddylated-Cul3-mediated ubiquitin ligation. The problem is that MG132 or MLN494 stabilizes CtIP in both KLHL15 overexpressing and not overexpressing cells (Figure 2b and 2c). In vitro it is clear (figure 2h). In vivo it will help if they repeat the overexpression of KLHL15 in cells depleted for Cul3.

We fully agree with the reviewer on this issue and have performed the suggested experiment. As shown in Figure 2c, siRNA-mediated depletion of CUL3 in KLHL15 overexpressing cells restores CtIP protein level, indicating that KLHL15-CUL3 is responsible for CtIP ubiquitination.

2. Does overexpression of CtIP behave like KHLH15 depleted or KO cells? Easy to test by comparing GFP and GFP-CtIP transfected cells in the presence of endogenous CtIP.

We have conducted the suggested experiment indicating that cells overexpressing CtIP do not behave like KLHL15-depleted cells in terms of DNA-end resection (Rebuttal Figure 3). However, this is not surprising as the two scenarios are quite different: GFP-CtIP-wt, even if overexpressed, can still be properly degraded by CUL3-KLHL15 whenever this is required to attenuate DNA-end resection. In a KLHL15 knockout background, or in cells expressing CtIP-Y842A, however, the CtIP protein turnover is misregulated, leading to the accumulation of CtIP in places where it should not be, thus causing the observed phenotypes.

Rebuttal Figure 3. U2OS Flp-In T-REx cells stably expressing doxycycline (Dox)-inducible GFP-CtIP-wt were cultivated in the absence (-) or presence of Dox (1 μg/ml) to induce GFP-CtIP expression. 48 h post induction, cells were either mock-treated or treated with CPT (1 μM) for 1 h. Total cell extracts were analysed by immunoblotting using the indicated antibodies.

3. Assays after DNA damage have been performed with a drug, CPT that causes DSBs in S phase. Thus, it repaired exclusively by recombination. What happens when the authors test other types of DNA damage that can be repaired by NHEJ also? Are cells overexpressing or depletion KHLH15 also affected the same way? Are they more sensitive to other types of DNA damage?

This is a very important issue and we addressed it using our different cell lines. In the original manuscript we showed that U2OS cells overexpressing KLHL15 are hypersensitive to CPT (Figure 4a). When we treat the same cells with ionizing radiation (IR), however, they are as resistance as the parental cells, even though, they are clearly resection-defective (Rebuttal Figure 4, left and middle panels). In fact, the same differential drug sensitivity was previously observed for CtIP-depleted cells (Sartori et al, 2007). However, given that CtIP-dependent resection, unlike the situation after CPT treatment, is mostly dispensable for the repair of IR-induced two-ended DSBs, we decided to exclude those data from the manuscript. Of course, if the referee feels that these findings are relevant for our work to be published, we are happy to include them in the manuscript.

Rebuttal Figure 4. Left panel, U2OS U2OS Flp-In T-REx cells inducibly expressing GFP-KLHL15-wt were cultivated in the presence or absence of Dox (1 μ g/ml). 24 h post-induction, cells were irradiated with the indicated amount of indicated ionizing radiation and survival was determined after ten days by colony-formation assay. Data are presented as the mean \pm SD ($n=2$). Middle panel, same cells as in **a** were irradiated at 10 Gy and 1 h later lysates were analysed by immunoblotting using the indicated antibodies. SE and LE denote short and long exposures of the same membrane, respectively. Right panel, HEK293Cas9/wt and HEK293Cas9/KLHL15 Δ cells were treated with the indicated doses of camptothecin and survival was determined after ten days by colony-formation assay. Data are presented as the mean \pm SD ($n=3$).

Next, we addressed the same issue in KLHL15 knocked cells and found that they are hypersensitive to IR but not to CPT (Figure 7g and Rebuttal Figure 4, right panel). This is very much in line with our previous finding demonstrating that cells lacking KLHL15 exhibit increased DNA-end resection channeling DSBs repair into HR and thus being defective in NHEJ (Figures 7f,h,i).

4. Related, I miss in several figures (4 and 7 mainly) a positive control, preferentially CtIP depletion. It will help to establish the degree in which the repair is compromised.

We apologize for this omission. We have now included new data, indicating that the resection phenotypes of CtIP-depleted cells follow the same trend but are stronger as compared to KLHL15-overexpressing cells (Supplementary Figures 4d-f). Moreover, we also include siCtIP as a positive control in Figures 7b-d.

5. CHX assays should be quantified. As the starting point is different, and Western blot signal saturate, it will help to strengthen the message.

To substantiate our findings, we have now quantified most of the CHX chase experiments and included them in the revised version (Supplementary Figures 1f, 2c and 7d; and Figure 3c). Please also refer to points 2,6,8 and 12 of Reviewer#1 for more details.

6. Figure 6c, a proper loading control (not Ponceau staining) is desirable.
We apologize for this but we believe that the Ponceau staining is the most important loading control to indicate that identical amounts of MBP-KLHL15 were used in every pull-down reaction.

7. Figure 8 does not add any information. The model is simple enough to be explained without a cartoon. I suggest to eliminate this figure.
We agree that the model is 'simple'. However, we believe that it would be favorable to retain it in the manuscript, as it gives the reader a concise and immediate understanding of the main findings and take-home message of our study. Therefore, we would like to keep it in the revised manuscript as Figure 9.

Reviewers' Comments:

Reviewer #1 (Remarks to the Author)

Reviewer #1:

In this manuscript the authors show that the cullin-3 E3 ligase substrate adaptor KLHL15 interacts with CtIP and targets it for ubiquitination and degradation. The major problems are summarized below:

1. Figure 1c shows that KLHL15 binds CtIP depending on both its BTB-BACK and Kelch domains. Does CtIP specifically bind KLHL15? The authors need to use other Cul3 substrate adaptors as control.

We think that our data is consistent with the idea that KLHL15 is indeed the substrate adaptor which specifically recruits CtIP to the CUL3 E3 Ligase complex. To further corroborate this conclusion, we have now included new data showing that KLHL22, another CUL3 substrate adaptor does not interact with CtIP (Supplementary Figure 1c).

To confirm that CtIP and Cul3-KLHL15 forms a trimeric complex, the authors should perform sequential IPs. Finally, the binding needs to be confirmed with endogenous proteins.

This is of course a very valid point and we made several efforts to address it. Nevertheless, we were unable to detect endogenous KLHL15-CUL3-CtIP complexes by conventional co-immunoprecipitation experiments. This could be explained by the fact that our anti-KLHL15 antibody was raised against an epitope mapping between amino acids 300-604. As this region encompasses the Kelch domain responsible for substrate binding, we concluded that the co-IP of endogenous CtIP mainly failed because the KLHL15 antibody masks the interaction site for CtIP. Similarly, the anti-CtIP antibodies at our disposal were raised against the C-terminal region of CtIP, again competing for KLHL15 binding. We also failed to co-precipitate endogenous CtIP using anti-CUL3 antibody, most likely because CUL3 interacts with various different substrate adaptors. Therefore, only a small proportion of CUL3 exists temporally in a complex with KLHL15-CtIP, which may be below detection levels. Finally, enzyme-substrate interactions are commonly known to be very transient and intrinsically difficult to detect by co-IPs.

However, we have included new results showing that endogenous KLHL15 and CUL3 are efficiently co-immunoprecipitated with GFP-CtIP-wt, but not with GFP-CtIP-Y842A (Figures 1b and 6c).

Moreover, we already showed that endogenous CtIP and CUL3 readily interact with FLAG-KLHL15 expressed in cells and now further substantiate these findings using two additional KLHL15 mutants (Figures 1d-f). Finally, we present new evidence that a short 15-mer CtIP peptide harboring the FRY KLHL15 docking motif efficiently pulls down endogenous CUL3-KLHL15 complex from parental HEK293 cells but not from HEK293 KLHL15 knockout cells (Fig. 5f).

Comments 1: Since the new results showed that a short 15-mer CtIP peptide harboring the FRY KLHL15 docking motif efficiently pulls down endogenous CUL3-KLHL15 complex from parental HEK293 cells, but not from HEK293 KLHL15 knockout cells, the statement that CtIP and Cul3-KLHL15 forms a trimetric complex is more solid.

2. In figure 1d, the N132A mutant lost the binding with Cul3, so, it should not mediate the degradation of CtIP. Why does the N132A mutant induce a decrease in CtIP protein levels? The half-life of CtIP should be measured after expression of WT and the N132A mutant.

We agree with the reviewer that data shown in Figure 1d is not fully in line with the concept that KLHL15-CUL3 targets CtIP for degradation. Therefore, we have repeated the co-IP experiment with FLAG-tagged KLHL15 '3-box' mutants (N132A and I136A) several times and realized that the interaction with CUL3 is not as strongly compromised as previously indicated. We apologize for this mistake and have now replaced the previous Figure with a more representative immunoblot (Figure 1e). This new data may also at least partially explain why KLHL15-N132A mutant is only partially defective in promoting CtIP degradation. As requested by the reviewer, we performed

several cycloheximide chase experiments and find that CtIP turnover rate is indeed comparable between KLHL15-N132A and KLHL15-wt, whereas KLHL15-Y552A significantly increased CtIP protein half-life (Rebuttal Figure 1 and Supplementary Figure 1f).

Rebuttal Figure 1. HEK293T cells transfected with either empty vector (EV) or the indicated FLAG-KLHL15 expression constructs for 48 h were treated with cycloheximide (CHX, 100 µg/ml) for the indicated time points and analysed by immunoblotting.

We have extensively discussed this contradicting issue with our collaborators (Matthias Peter and Radoslav Enchev) and we think that it could be at least in part explained by the molecular architecture of CUL3-based E3 ligase complexes. It is known that most, if not all, CUL3 ligases dimerize through the BTB domain of their adaptor protein and it is proposed that those dimers are engaged in a single complex (Merlet et. al, 2009 and Canning et. al, 2013). The best investigated example of such dimerization is CUL3-Keap1-Nrf2, in which two Keap1 substrate adaptor proteins simultaneously bind to a single Nrf2 target molecule, thus building a complex consisting of two CUL3, two Keap1 and one Nrf2 (Canning et al, 2015). We believe that the same bivalent model can be applied to CUL3-KLHL15-CtIP, in which two KLHL15 subunits are required for the interaction with CtIP. If this is the case, then KLHL15 3-box mutants (N132A and I136A) retain the capacity to heterodimerize with endogenous wild-type KLHL15 (present in our cells), which in turns interacts with CUL3 and ensures the formation of a functional catalytic center still able to ubiquitinate CtIP. In contrast, as both KLHL15 subunits are required for CtIP binding, Kelch repeat domain mutants (G386C or Y552A) more potently abolish the interaction with CtIP and ultimately, the degradation of CtIP (as indicated in Figure 1f and Supplementary Figure 1f). However, further detailed structural investigations are needed to experimentally support this model, which in our view are beyond the scope of this study.

Comments 2: The authors showed that KLHL15 N132A mutant completely lost the binding with Cul3 comparing to KLHL15 wildtype, while currently showed that N132A mutant could still bind Cul3. How reproducible is this result?

3. In figure 2a, the Y552A mutant is expressed less than the WT, so, it is difficult to make a conclusion on its activity. Again, the half-life of CtIP should be measured after expression of WT and the Y552A mutant.

We have analyzed the half-life of endogenous CtIP in U2OS cells and compared it to cells expressing either KLHL15-wt or Y552A (Rebuttal Figure 2). However, CHX treatment of U2OS cells does not lead to a reduction in CtIP protein levels. We reasoned that this is most likely due to very low KLHL15 expression levels in U2OS, thereby rendering CtIP much more stable as compared to the situation in HEK293(T) cells (compared to point 4). Nevertheless, when we induced the expression of KLHL15 in the stable U2OS cells, CtIP protein half-life is increased in Y552A compared to wild-type cells (Rebuttal Figure 2).

Rebuttal Figure 2. U2OS Flp-In T-REx cells inducibly expressing GFP-KLHL15-wt or GFP-KLHL15-Y552A were cultivated in presence or absence of Dox (1 µg/ml). 24 h post-induction cells were treated with cycloheximide (CHX, 100 µg/ml) for the indicated time points and analysed by immunoblotting.

For these reasons and for clarity, we have decided to exclude this experiment from the revised manuscript. However, as already outlined above, we performed CHX chase experiments and measured CtIP half-life in HEK293T cells transiently transfected with KLHL15-wt and KLHL15-Y552A (Supplementary Figure 1f). Lastly, we decided to exclude Figure 2a from the revised manuscript, as we have now more extensively characterized U2OS GFP-KLHL15-Y552A stable clones in the revised manuscript (Figure 4 and Supplementary Figures 4b-c). Concerning KLHL15-Y552A expression levels in stable U2OS cells, please also refer to point 9.

Comments 3: The authors showed that CtIP protein levels is very stable in U2OS cells and explained that "this is most likely due to very low KLHL15 expression levels in U2OS", suggesting that KLHL15 mediated CtIP degradation is difficult to detect under basal level. If this is the case, why is there a strong accumulation of CtIP protein levels caused by silencing of KLHL15 in U2OS cells (Figure 2d)?

4. Since an antibody for WB is available, in figure 2d, 2e and 2f, the authors should show KLHL15 protein levels to confirm that the siRNA to KLHL15 worked.

We apologize for these omissions. As shown in Figure 2d, we could confirm the knockdown of KLHL15 by western blot analysis. However, since the amount of endogenous KLHL15 is extremely low in U2OS cells, we could only detect KLHL15 in the chromatin-enriched fraction. For the same reason, we are unable to show KLHL15 immunoblot in Figure 2e. However, we believe that the GFP immunoblot is sufficient to verify efficient silencing of KLHL15 expression by siRNA oligos. Finally, concerning KLHL15 protein levels in Figure 2f (moved to Supplementary Figure S2c in the revised manuscript), we now include a KLHL15 immunoblot confirming partial downregulation of KLHL15 in HEK293T cells.

Comments 4: If the KLHL15 expression is extremely low in U2OS cells, the authors should use other cell lines to study the role of endogenous KLHL15.

5. Figure 2d: The siRNA #3 is much less efficient than #1. However, #3 siRNA induces the same effect on the CtIP protein level, as #1. How is this possible?

The simplest way for us to answer this valid question is that KLHL15 protein levels are not proportional to KLHL15 mRNA transcript levels, as measured in this experiment (moved to Supplementary Figure 2A). In other words, KLHL15 mRNA turnover may be much shorter than KLHL15 protein turnover and, although mRNA levels upon siRNA#1 transfection are approximately 2-fold lower than upon siRNA#3 transfection, KLHL15 protein levels are similarly reduced with both siRNA oligos. In fact, our new western blot data reveals that the all KLHL15 siRNA oligos used are downregulating KLHL15 (Figure 2d). Still, we agree that there is more residual KLHL15 protein left with siRNA#3. Perhaps, because of the dimeric structure of CRL3s (see point 1 above), this amount of KLHL15 is still not sufficient to promote efficient CtIP ubiquitination and proteasomal degradation.

Comments 5: the response is not clear.

6. Figure 2f: KLHL15 siRNA dramatically increased the protein level of CtIP, however, the half-life is only slightly increased. How reproducible is this result? The authors should quantify the half-life in several experiments.

This is an important point. We have now independently reproduced this CHX chase experiment several times and included the quantification data in a new Supplementary Figure 2c.

Comments 6: OK.

7. Figure 2g and 2h showed that KLHL15 promoted the ubiquitination of CtIP in vivo and in vitro, is it formed K48 ubiquitin chain? A negative control with another ubiquitin ligase would be advisable.

We thank the reviewer for this comment. By using ubiquitin mutants (K48 only and K48R) we now report that CUL3-KLHL15 forms K48-linked ubiquitin chains on CtIP (Supplementary Figure 2e). Moreover, in the same experiment, we included CUL3-KLHL22 E3 ubiquitin ligase as a negative control (Supplementary Figure 2e).

Comments 7: OK.

8. Figure 3c: The authors need to show a long and short exposure of the CtIP blot that allows a

direct comparison of the effect on the half-life.

We fully agree with the reviewer and modified Figure 3c accordingly. In addition, we have repeated this experiment multiple times and included the quantification data in the same Figure.

Comments 8: OK.

9. Figure 4b: Again, the Y552A mutant expression level is much less than the WT, so, the phenotype might be caused by the different expression level of KLHL15, not the Cul3 binding. We fully agree that this is an important issue. Therefore, we have now included a western blot analysis of three separate stable U2OS clones inducibly expressing GFP-KLHL15-Y552A, clearly demonstrating that regardless of the respective Y552A expression levels, CtIP protein stability and, thus, DNA-end resection activity remains largely unaffected.

Comments 9: OK.

10. In figure 5b, the authors showed that delta C1 mutant lost the binding with KLHL15. This could be due to a change in localization (the C-terminal domain is required for DNA-end resection) and needs to be verified.

This is indeed possible. Yet, it was previously shown that GFP-CtIP Δ C1 properly localizes in the nucleus (Sartori et al, 2007). Moreover, we show that full-length GFP-CtIP harboring a single amino acid substitution in the C-terminal domain (Y842A) is defective in KLHL15 interaction, but is still properly localized in the nucleus and efficiently recruited to sites of DNA damage (Supplementary Figures 7b-c).

Comments 10: OK.

11. Fig. 6d: the mutants seem less ubiquitinated, but also less precipitated (i.e. less protein is pulled down), which could be the real reason for the apparent decrease in ubiquitination. The experiment needs to be normalized.

In this special type of ubiquitination assay, cells are transfected with His-tagged ubiquitin. Subsequently, ubiquitinated proteins are enriched using a Ni-NTA resin. Ni-NTA eluates are then analyzed by western blot with an antibody against the protein of interest, in this case GFP-CtIP. Therefore, GFP-CtIP-Y842A is indeed less precipitated because it is less ubiquitinated, even though its expression level is comparable to GFP-CtIP-wt as shown in the input blot (Figure 6f).

Comments 11: OK.

12. Figure 6e: Although the Y842A mutant expresses more than the WT when co-expressed with KLHL15, it doesn't mean that the Y842A mutant is more stable. A measurement of the half-life is essential.

We completely agree with the reviewer on this point and thus measured the half-life of GFP-CtIP-wt and Y842A in several independent experiments. The western blot and quantification data clearly indicate that CtIP-Y842A has a prolonged half-life compared to CtIP-wt (Supplementary Figure 7d).

Comments 12: OK

Reviewer #2 (Remarks to the Author)

The original manuscript from Ferreti et al was already of a very high technical and scientific quality. This revised version is even stronger. The authors have successfully address all my comments, and even we disagree in a couple of minor thing they are not important enough to preclude publication. In terms of how solid is the science behind the paper I have no doubts. My

main concern was that a study focusing on the normal turnover of a specific protein might not be of interest for a general audience and more suitable for a specialist. The authors have made an effort to link their observations with a regulatory loop through PIN1 with partial success. Although I still have some doubts about how appropriate is the main message for Nat Communication, I do appreciate the authors efforts and I also understand this is a very subjective matter. In that regard, I think it might be a decision better judged by the editor than by a referee and publication of the paper should not be blocked by a subjective criterion.

Response to referees:

We would like to thank both reviewers for reporting on our revised manuscript. Please find below our point-by-point response to the few remaining comments.

Reviewer #1:

In this manuscript the authors show that the cullin-3 E3 ligase substrate adaptor KLHL15 interacts with CtIP and targets it for ubiquitination and degradation. The major problems are summarized below:

1. Figure 1c shows that KLHL15 binds CtIP depending on both its BTB-BACK and Kelch domains. Does CtIP specifically bind KLHL15? The authors need to use other Cul3 substrate adaptors as control.

We think that our data is consistent with the idea that KLHL15 is indeed the substrate adaptor which specifically recruits CtIP to the CRL3 E3 Ligase complex. To further corroborate this conclusion, we have now included new data showing that KLHL22, another CUL3 substrate adaptor does not interact with CtIP (Supplementary Figure 1c).

To confirm that CtIP and Cul3-KLHL15 forms a trimeric complex, the authors should perform sequential IPs. Finally, the binding needs to be confirmed with endogenous proteins.

This is of course a very valid point and we made several efforts to address it. Nevertheless, we were unable to detect endogenous KLHL15-CUL3-CtIP complexes by conventional co-immunoprecipitation experiments. This could be explained by the fact that our anti-KLHL15 antibody was raised against an epitope mapping between amino acids 300-604. As this region encompasses the Kelch domain responsible for substrate binding, we concluded that the co-IP of endogenous CtIP mainly failed because the KLHL15 antibody masks the interaction site for CtIP. Similarly, the anti-CtIP antibodies at our disposal were raised against the C-terminal region of CtIP, again competing for KLHL15 binding. We also failed to co-precipitate endogenous CtIP using anti-CUL3 antibody, most likely because CUL3 interacts with various different substrate adaptors. Therefore, only a small proportion of CUL3 exists temporally in a complex with KLHL15-CtIP, which may be below detection levels. Finally, enzyme-substrate interactions are commonly known to be very transient and intrinsically difficult to detect by co-IPs.

However, we have included new results showing that endogenous KLHL15 and CUL3 are efficiently co-immunoprecipitated with GFP-CtIP-wt, but not with GFP-CtIP-Y842A (Figures 1b and 6c). Moreover, we already showed that endogenous CtIP and CUL3 readily interact with FLAG-KLHL15 expressed in cells and now further substantiate these findings using two additional KLHL15 mutants (Figures 1d-f). Finally, we present new evidence that a short 15-mer CtIP peptide harboring the FRY KLHL15 docking motif efficiently pulls down endogenous CUL3-KLHL15 complex from parental HEK293 cells but not from HEK293 KLHL15 knockout cells (Fig. 5f).

Comments 1: Since the new results showed that a short 15-mer CtIP peptide harboring the FRY KLHL15 docking motif efficiently pulls down endogenous CUL3-KLHL15 complex from parental HEK293 cells, but not from HEK293 KLHL15 knockout cells, the statement that CtIP and Cul3-KLHL15 forms a trimetric complex is more solid.

Response: We thank the reviewer for appreciating our efforts in strengthening the concept that CtIP and CUL3-KLHL15 form a trimeric protein complex.

2. In figure 1d, the N132A mutant lost the binding with Cul3, so, it should not mediate the degradation of CtIP. Why does the N132A mutant induce a decrease in CtIP protein levels? The half-life of CtIP should be measured after expression of WT and the N132A mutant.

We agree with the reviewer that data shown in Figure 1d is not fully in line with the concept that KLHL15-CUL3 targets CtIP for degradation. Therefore, we have repeated the co-IP experiment with FLAG-tagged KLHL15 '3-box' mutants (N132A and I136A) several times and realized that the interaction with CUL3 is not as strongly compromised as previously indicated. We apologize for this mistake and have now replaced the previous Figure with a more representative immunoblot (Figure 1e). This new data may also at least partially explain why KLHL15-N132A mutant is only partially defective in promoting CtIP degradation. As requested by the reviewer, we performed several cycloheximide chase experiments and find that CtIP turnover rate is indeed comparable between KLHL15-N132A and KLHL15-wt, whereas KLHL15-Y552A significantly increased CtIP protein half-life (Rebuttal Figure 1 and Supplementary Figure 1f).

Rebuttal Figure 1. HEK293T cells transfected with either empty vector (EV) or the indicated FLAG-KLHL15 expression constructs for 48 h were treated with cycloheximide (CHX, 100 µg/ml) for the indicated time points and analysed by immunoblotting.

We have extensively discussed this contradicting issue with our collaborators (Matthias Peter and Radoslav Enchev) and we think that it could be at least in part explained by the molecular architecture of CUL3-based E3 ligase complexes. It is known that most, if not all, CUL3 ligases dimerize through the BTB domain of their adaptor protein and it is proposed that those dimers are engaged in a single complex (Merlet et. al, 2009 and Canning et. al, 2013). The best investigated example of such dimerization is CUL3-Keap1-Nrf2, in which two Keap1 substrate adaptor proteins simultaneously bind to a single Nrf2 target molecule, thus building a complex consisting of two CUL3, two

Keap1 and one Nrf2 (Canning et al, 2015). We believe that the same bivalent model can be applied to CUL3-KLHL15-CtIP, in which two KLHL15 subunits are required for the interaction with CtIP. If this is the case, then KLHL15 3-box mutants (N132A and I136A) retain the capacity to heterodimerize with endogenous wild-type KLHL15 (present in our cells), which in turns interacts with CUL3 and ensures the formation of a functional catalytic center still able to ubiquitinate CtIP. In contrast, as both KLHL15 subunits are required for CtIP binding, Kelch repeat domain mutants (G386C or Y552A) more potently abolish the interaction with CtIP and ultimately, the degradation of CtIP (as indicated in Figure 1f and Supplementary Figure 1f). However, further detailed structural investigations are needed to experimentally support this model, which in our view are beyond the scope of this study.

Comments 2: *The authors showed that KLHL15 N132A mutant completely lost the binding with Cul3 comparing to KLHL15 wildtype, while currently showed that N132A mutant could still bind Cul3. How reproducible is this result?*

Response: *First of all, we apologize to both reviewers for having included a partially 'misleading' figure regarding the interaction of the N132A CUL3-Box mutant of KLHL15 with CUL3 in the first version of our manuscript. Therefore, in the revised version of the manuscript, we have included another CUL3-Box mutant of KLHL15 (I136A) showing that both N132A and I136A are still able to interact with neddylated (=activated) CUL3, albeit with reduced efficiency compared to wild-type KLHL15 (Figure 1e). In addition, we include here two independent experiments confirming our previous observation that KLHL15-N132A is only partially defective in CUL3 interaction (see **Rebuttal Figure 1**). We hope that this data 'convinces' this reviewer about the reproducibility of our previous result. We also agree with reviewer that the effect of KLHL15-N132A is rather mild with regards to CUL3 interaction (the Kelch repeat mutants Y552A and G386C show a similar CUL3-interaction defect as compared to N132A, see Rebuttal Figure). Therefore, we decided to focus on the Kelch repeat mutant (Y552A) for the remainder of our study, because it displayed a more 'dominant' effect on CtIP protein stability compared to N132A.*

Rebuttal Figure 1: (a,b), HEK293T cells were transfected with either empty vector (EV) or the indicated FLAG-KLHL15 expression constructs. 48 hours after transfection, cells were lysed and whole-cell extracts were subjected to immunoprecipitation (IP) using anti-FLAG M2 affinity resin (lower panel). Inputs and recovered protein complexes were analysed by immunoblotting. Asterisks indicate neddylated CUL3.

3. In figure 2a, the Y552A mutant is expressed less than the WT, so, it is difficult to make a conclusion on its activity. Again, the half-life of CtIP should be measured after expression of WT and the Y552A mutant.

We have analyzed the half-life of endogenous CtIP in U2OS cells and compared it to cells expressing either KLHL15-wt or Y552A (Rebuttal Figure 2). However, CHX treatment of U2OS cells does not lead to a reduction in CtIP protein levels. We reasoned that this is most likely due to very low KLHL15 expression levels in U2OS, thereby rendering CtIP much more stable as compared to the situation in HEK293(T) cells (compared to point 4). Nevertheless, when we induced the expression of KLHL15 in the stable U2OS cells, CtIP protein half-life is increased in Y552A compared to wild-type cells (Rebuttal Figure 2).

Rebuttal Figure 2. U2OS Flp-In T-REx cells inducibly expressing GFP-KLHL15-wt or GFP-KLHL15-Y552A were cultivated in presence or absence of Dox (1 µg/ml). 24 h post-induction cells were treated with cycloheximide (CHX, 100 µg/ml) for the indicated time points and analysed by immunoblotting.

For these reasons and for clarity, we have decided to exclude this experiment from the revised manuscript. However, as already outlined above, we performed CHX chase experiments and measured CtIP half-life in HEK293T cells transiently transfected with KLHL15-wt and KLHL15-Y552A (Supplementary Figure 1f). Lastly, we decided to exclude Figure 2a from the revised manuscript, as we have now more extensively characterized U2OS GFP-KLHL15-Y552A stable clones in the revised manuscript (Figure 4 and Supplementary Figures 4b-c). Concerning KLHL15-Y552A expression levels in stable U2OS cells, please also refer to point 9.

Comments 3: *The authors showed that CtIP protein levels is very stable in U2OS cells and explained that "this is most likely due to very low KLHL15 expression levels in U2OS", suggesting that KLHL15 mediated CtIP degradation is difficult to detect under basal level. If this is the case, why is there a strong accumulation of CtIP protein levels caused by silencing of KLHL15 in U2OS cells (Figure 2d)?*

Response: *We agree with the reviewer that our data suggest a prolonged half-life of CtIP in U2OS cells as compared to HEK293 cells. Nevertheless, and similar to HEK293 cells, we find that CtIP protein levels in U2OS cells increased upon downregulation of KLHL15. However, we repeatedly noted that KLHL15 protein levels are generally higher in HEK293 cells as compared*

to U2OS cells, where we could only readily detect KLHL15 in chromatin-enriched fractions, suggesting that relative KLHL15 expression levels may contribute to the specific turnover rate of CtIP. Thus, the potent accumulation of CtIP observed in U2OS upon KLHL15 depletion could be explained by the fact that KLHL15-mediated turnover control of CtIP exists also in U2OS but at a lower rate, thereby giving rise to CtIP accumulation as observed in Figure 2d. Moreover, in the experiment shown in Figure 2d, KLHL15 is silenced for 48 hours, whereas the CHX pulse-chase experiments shown in Rebuttal Figure 2 are performed in presence of KLHL15 (lanes 1-4) and for short time points (up to 5 hours).

4. Since an antibody for WB is available, in figure 2d, 2e and 2f, the authors should show KLHL15 protein levels to confirm that the siRNA to KLHL15 worked.

We apologize for these omissions. As shown in Figure 2d, we could confirm the knockdown of KLHL15 by western blot analysis. However, since the amount of endogenous KLHL15 is extremely low in U2OS cells, we could only detect KLHL15 in the chromatin-enriched fraction. For the same reason, we are unable to show KLHL15 immunoblot in Figure 2e. However, we believe that the GFP immunoblot is sufficient to verify efficient silencing of KLHL15 expression by siRNA oligos. Finally, concerning KLHL15 protein levels in Figure 2f (moved to Supplementary Figure S2c in the revised manuscript), we now include a KLHL15 immunoblot confirming partial downregulation of KLHL15 in HEK293T cells.

Comments 4: If the KLHL15 expression is extremely low in U2OS cells, the authors should use other cell lines to study the role of endogenous KLHL15.

Response: In fact, due to of low KLHL15 expression levels in U2OS, we have used HEK293 cells to knockout KLHL15 using CRISPR-Cas9 and 'to the study the role of endogenous KLHL15'. We have previously investigated the effect of KLHL15 downregulation in HeLa and RPE-1 cells, showing that CtIP levels increase after KLHL15 silencing (**Rebuttal Figure 2**). Moreover, like in U2OS cells, we only detect KLHL15 in the chromatin-enriched fraction of HeLa cells, suggesting that KLHL15 expression may be generally lower in cancer cell lines. Taken together, our data suggests that CtIP maybe more stable in U2OS and HeLa cells (lower KLHL15 expression levels) as compared to HEK293 (higher KLHL15 expression levels) and RPE-1 cells but that KLHL15 is a global mediator of CtIP proteasomal degradation independently of the cell type.

Rebuttal 2 Figure 2: **a**, HeLa cells were transfected with the indicated siRNA oligos. 48 h post siRNA transfection, cells were either mock-treated or treated with camptothecin (1 μ M) for 1 h. Total cell extracts and chromatin fractions were analysed by immunoblotting using the indicated antibodies. **b**, U2OS or RPE1 cells were transfected with the indicated siRNA oligos. 48 h post siRNA transfection, cells were cultured either in normoxia (N, 20% O₂) or hypoxia (H, 0.2% O₂) for 8 h. Whole-cell lysates were analysed by immunoblotting using the indicated antibodies. Of note: unfortunately, the KLHL15 antibody was not available to us at the time of this experiment.

5. Figure 2d: The siRNA #3 is much less efficient than #1. However, #3 siRNA induces the same effect on the CtIP protein level, as #1. How is this possible? The simplest way for us to answer this valid question is that KLHL15 protein levels are not proportional to KLHL15 mRNA transcript levels, as measured in this experiment (moved to Supplementary Figure 2A). In other words, KLHL15 mRNA turnover may be much shorter than KLHL15 protein turnover and, although mRNA levels upon siRNA#1 transfection are approximately 2-fold lower than upon siRNA#3 transfection, KLHL15 protein levels are similarly reduced with both siRNA oligos. In fact, our new western blot data reveals that the all KLHL15 siRNA oligos used are downregulating KLHL15 (Figure 2d). Still, we agree that there is more residual KLHL15 protein left with siRNA#3. Perhaps, because of the dimeric structure of CRL3s (see point 1 above), this amount of KLHL15 is still not sufficient to promote efficient CtIP ubiquitination and proteasomal degradation.

Comments 5: *the response is not clear.*

Response: *We are sorry for not being able to giving a more clear answer to this issue. A simple explanation could be that the knockdown of KLHL15 induced by KLHL15 siRNA#3, even tough less efficient as compared to #1, is sufficient to block CtIP proteasomal degradation. In other words, the remaining KLHL15 protein is not sufficient to form active CUL3-KLHL15 E3 ligase complexes (according to the bivalent model explained above in our response to Comment 1) in U2OS cells, which already have lower KLHL15 basal levels (see our response to Comments 3 and 4).*

Reviewer #2:

The original manuscript from Ferreti et al was already of a very high technical and scientific quality. This revised version is even stronger. The authors have successfully address all my comments, and even we disagree in a couple of minor thing they are not important enough to preclude publication. In terms of how solid is the science behind the paper I have no doubts.

My main concern was that a study focusing on the normal turnover of a specific protein might not be of interest for a general audience and mores suitable for an specialist. The authors have made an effort to link their observations with a regulatory loop through PIN1 with partial success. Although I still have some doubts about how appropriate is the main message for Nat Communication, I do appreciate the authors efforts and I also understand this is a very subjective matter. In that regard, I think it might be a decision better judged by the editor than by a referee and publication of the paper should not be blocked by a subjective criterion.

Response: *We thank this reviewer for his/her positive comments on the revised version of our manuscript.*

We are obviously of the opinion that in-depth knowledge on the multi-faceted regulation systems for CtIP protein function is of interest for both specialists (e.g. those active in the Cullin and E3 Ligase fields) and a general audience. The latter may for example apply to recent findings showing that alternative end-joining (alt-EJ) fuels genomic instability, chromosomal translocations and, ultimately cancer progression. Since CtIP-dependent resection of DNA double-strand breaks (DSBs) is a prerequisite for alt-EJ (as it is also for error-free homologous recombination), pathological conditions leading to the accumulation of CtIP protein levels (e.g. KLHL15 mutations or KLHL15 silencing) may promote increased frequency of alt-EJ mediated DSB repair, specifically in a context where homologous recombination is not functional (such as in BRCA1/2-mutated cancers).

Reviewers' Comments:

Reviewer #1 (Remarks to the Author)

I have examined the revised manuscript. I think that the authors have correctly addressed my comments and changed the manuscript accordingly.